# XLLC-Net: A lightweight and explainable CNN for accurate lung cancer classification using histopathological images

**Jamin Rahman Jim**[ID]☯, **Md. Eshmam Rayed**[ID]☯, **M.F. Mridha**[ID]*, **Kamruddin Nur**[ID]

Department of Computer Science, American International University-Bangladesh, Dhaka 1229, Bangladesh

☯ These authors contributed equally to this work.

* firoz.mridha@aiub.edu

**Data availability statement:** https://github.com/tampapath/lung_colon_image_set?tab=readme-ov-file

## Abstract

Lung cancer imaging plays a crucial role in early diagnosis and treatment, where machine learning and deep learning have significantly advanced the accuracy and efficiency of disease classification. This study introduces the Explainable and Lightweight Lung Cancer Net (XLLC-Net), a streamlined convolutional neural network designed for classifying lung cancer from histopathological images. Using the LC25000 dataset, which includes three lung cancer classes and two colon cancer classes, we focused solely on the three lung cancer classes for this study. XLLC-Net effectively discerns complex disease patterns within these classes. The model consists of four convolutional layers and contains merely 3 million parameters, considerably reducing its computational footprint compared to existing deep learning models. This compact architecture facilitates efficient training, completing each epoch in just 60 seconds. Remarkably, XLLC-Net achieves a classification accuracy of 99.62% ± 0.16%, with precision, recall, and F1 score of 99.33% ± 0.30%, 99.67% ± 0.30%, and 99.70% ± 0.30%, respectively. Furthermore, the integration of Explainable AI techniques, such as Saliency Map and GRAD-CAM, enhances the interpretability of the model, offering clear visual insights into its decision-making process. Our results underscore the potential of lightweight DL models in medical imaging, providing high accuracy and rapid training while ensuring model transparency and reliability.

## 1 Introduction

Lung cancer remains a leading cause of cancer-related mortality worldwide, highlights the critical need for early and accurate diagnostic methods to enhance patient survival [1]. Central to lung cancer diagnosis is histopathological imaging, which involves microscopic examination of tissue samples to identify malignant cells. However, this manual analysis process is labor-intensive, susceptible to human error, and often yields inconsistent results [2]. The recent integration of machine learning (ML) and deep learning (DL) into medical imaging has brought about a transformative shift, providing robust tools capable of analyzing intricate image patterns with remarkable precision and reliability. These technological

**Funding:** The author(s) received no specific funding for this work.

**Competing interests:** The authors have declared that no competing interests exist.

advancements enable the automation of lung cancer classification, significantly accelerating the diagnostic workflow, reducing the burden on pathologists, and improving diagnostic accuracy [3]. Consequently, this streamlining of the diagnostic process has the potential to enhance patient outcomes by facilitating earlier and more reliable detection of lung cancer.

The application of ML and DL models in lung cancer imaging has yielded exceptional accuracy and reliability, often outperforming traditional diagnostic methods [4]. For instance, DL models can identify minute abnormalities in histopathological images that might be overlooked by human eyes, leading to earlier and more precise diagnoses [5]. Despite these advancements, significant challenges impede their widespread clinical adoption [6]. Many state-of-the-art DL models are highly computationally intensive, requiring substantial hardware resources such as GPUs or specialized AI accelerators, and lengthy training times, which can be prohibitive for many healthcare facilities [7]. Furthermore, the "black-box" nature of these models—where the decision-making process is opaque—poses a substantial barrier [8]. Clinicians and pathologists may struggle to trust and accept recommendations from a model whose internal workings and reasoning are not transparent [9]. For example, a model might accurately classify a tumor as malignant, but without understanding which features or patterns the model used to arrive at that conclusion, healthcare professionals may be hesitant to rely on it fully. Therefore, addressing these issues, such as by developing more efficient models and incorporating explainable AI (XAI) techniques, is crucial for the practical implementation and acceptance of DL models in clinical environments.

Convolutional neural networks (CNNs) have demonstrated outstanding performance in image classification tasks due to their ability to learn and utilize hierarchical feature representations, which enable them to detect and interpret intricate patterns crucial for accurate medical diagnoses [10]. In the realm of medical image analysis, CNN-based models excel by identifying subtle features in images that may be pivotal for diagnosing conditions such as lung cancer. Lightweight CNNs, engineered with fewer parameters and layers, provide significant advantages by reducing computational demands and shortening training times while maintaining high-performance levels. These streamlined models are particularly advantageous in clinical settings where computational resources might be limited, making them more practical for real-world applications [11]. Furthermore, the integration of XAI techniques, such as Saliency Maps and Gradient-weighted Class Activation Mapping (GRAD-CAM), enhances the transparency of these models [12]. By visually highlighting the specific regions of an image that influence the model's decisions, XAI techniques allow clinicians to understand and trust the model's reasoning processes. This transparency is vital for fostering clinical confidence and encouraging the adoption of AI tools in healthcare, ensuring that these advanced technologies can be effectively integrated into routine medical practice.

Given the ongoing demand for efficient, accurate, and interpretable models in medical imaging, this paper presents the Explainable and Lightweight Lung Cancer Net (XLLC-Net). XLLC-Net is a CNN meticulously crafted for the classification of lung cancer using histopathological images. It effectively balances the dual imperatives of delivering high performance and addressing the practical constraints typical of clinical environments. Specifically, XLLC-Net is designed to operate with minimal computational resources while maintaining exceptional accuracy, making it highly suitable for real-world medical applications. By integrating advanced yet lightweight architecture with XAI components, our research aims to close the gap between sophisticated DL models and their practical deployment in healthcare settings. We illustrate that with thoughtful design, a lightweight neural network can achieve

top-tier results, offering a viable solution to the computational challenges and interpretability concerns that often hinder the clinical adoption of advanced DL models.

The main contributions of our paper are as follows:

- **Introduction of XLLC-Net:** We present a novel lightweight CNN with only 3 million parameters, optimized for rapid training and high accuracy in lung cancer classification.
- **Integration of XAI:** We incorporate Saliency Maps and GRAD-CAM to enhance the interpretability of XLLC-Net, allowing clinicians to visualize and understand the critical regions influencing the model's decisions. This transparency fosters greater trust and facilitates the practical adoption of AI tools in medical diagnostics.
- **Performance Metrics:** Our model achieves a classification accuracy of 99.62% ± 0.16%, with precision, recall, and F1 scores of 99.33% ± 0.30%, 99.67% ± 0.30%, and 99.70% ± 0.30%, respectively, demonstrating its robustness and effectiveness. The performance and efficiency of XLLC-Net are rigorously validated using the LC25000 dataset, which includes diverse classes of lung and colon cancer, ensuring comprehensive evaluation and generalizability.

The rest of the paper is structured as follows: Section 2 explores the related work, and then Section 3 discusses the methodology of our experiment. After that Section 4 analyzes the experimental results and Section 5 explains the limitations and discusses the future research opportunities for our proposed model. Finally, Section 6 concludes the paper.

## 2 Related work

The application of DL in medical imaging has experienced significant advancements, particularly in the classification and diagnosis of complex diseases such as lung cancer. DL models have demonstrated exceptional proficiency in analyzing intricate histopathological images, often surpassing traditional diagnostic methods in accuracy and reliability. Esteva et al. showcased the transformative potential of DL in dermatology by achieving classification accuracies comparable to those of expert dermatologists [13]. In the context of lung cancer detection, DL models have been instrumental in automatically identifying malignant patterns within histopathological images with remarkable precision, as highlighted by Litjens et al. [11]. For instance, Autoencoders have been utilized for feature extraction and dimensionality reduction, enabling efficient analysis of large histopathological datasets [14]. Recurrent neural networks (RNNs), particularly long short-term memory (LSTM) networks, have been explored for their capability to handle sequential data and temporal dependencies in medical imaging [15]. Additionally, generative adversarial networks (GANs) have shown promise in augmenting histopathological datasets by generating synthetic yet realistic images, thereby addressing the challenge of limited annotated data [16]. However, despite these successes, DL models are often criticized for their high computational requirements and the opacity of their decision-making processes [17].

CNNs have revolutionized lung cancer imaging by significantly enhancing diagnostic accuracy and efficiency [18]. Their ability to automatically learn and extract intricate features from histopathological images allows for the precise identification of cancerous tissues. CNNs reduce subjective variability and errors in manual interpretation, standardizing diagnostic processes and leading to more reliable outcomes. These advancements facilitate early detection and treatment planning, ultimately improving patient prognosis and survival rates. By automating complex image analysis tasks, CNNs have become a critical tool in modern medical diagnostics, advancing lung cancer detection and management. For instance, Atiya et

al. used a dual-state transfer learning method with deep CNNs to significantly enhance the accuracy of classifying and detecting non-small cell lung cancers in CT scans. The ResNet50 model, in particular, achieved high accuracy rates of 94% during training, 92.57% during validation, and 96.12% during testing [19]. In another work, Hamed et al. proposed a CNN-based model that effectively classifies lung cancer histopathology images as benign or malignant with an accuracy of 99.3% to 99.8% using the LC25000 dataset. The study demonstrates that the model performs comparably with state-of-the-art methods, significantly aiding in the early detection and diagnosis of lung cancer [20]. Furthermore, Perez et al. presented a highly effective framework for automated lung cancer diagnosis using LDCT scans, achieving a 99.6% recall in nodule detection and a precision increase by a factor of 2000 through a three-dimensional CNN. The cancer predictor demonstrated strong performance with an ROC AUC of 91.3%, ranking 1st in the ISBI 2018 Lung Nodule Malignancy Prediction challenge [21]. Despite these advancements, CNNs often require large datasets for training and significant computational resources, which can be a barrier to their widespread adoption in clinical settings.

To mitigate the limitations of training from scratch, pre-trained models have been increasingly utilized in medical image analysis. Notable pre-trained models such as AlexNet [22], ResNet50 [23], VGG16 [24], and VGG19 [25] have been extensively employed for lung cancer imaging tasks. These models, initially trained on large-scale datasets, can be fine-tuned to specific medical applications, thereby reducing the need for extensive labeled medical datasets. Despite their success, pre-trained models still face challenges related to computational efficiency and resource requirements, which can limit their practical deployment in clinical settings. The complexity and size of these models necessitate significant computational power, highlighting the ongoing need for more efficient and lightweight solutions that maintain high diagnostic performance. Therefore, the need for customized CNN models tailored to specific medical imaging tasks has led to the development of several specialized architectures. These models aim to balance accuracy and computational efficiency, addressing the constraints faced by pre-trained models. Cibi et al. presented a customized deep CNN model using Capsule Networks (CapsNet), an advanced neural network architecture designed to preserve spatial relationships between features through dynamic routing, for classifying cervical cancer stages from MR images, achieving an accuracy of 90.28% [26]. The results indicate that the CNN-CapsNet model can significantly improve diagnosis accuracy and reliability, aiding treatment planning in healthcare. In another work, Faruqui et al. proposed a 22-layer hybrid deep-CNN model, which enhances lung cancer diagnosis by combining CT scan images and wearable sensor-based MIoT data, achieving high accuracy (96.81%) and low false positive rate (3.35%). It also effectively classifies early-stage lung cancers into subclasses with 91.6% accuracy, demonstrating superior performance over similar models [27]. Nevertheless, while these models reduce computational requirements, they often still lack the transparency needed for clinical trust and acceptance.

XAI techniques have been introduced to address the black-box nature of DL models, providing insights into the model's decision-making process. Methods such as Saliency Maps [28] and GRAD-CAM [29] highlight the regions of an image that contribute most to the model's predictions. These techniques are crucial in medical imaging, where understanding the rationale behind a diagnosis is as important as the diagnosis itself. Despite the progress, the integration of XAI into lightweight and efficient models has been limited, leaving a gap in the development of models that are both interpretable and practical for clinical use. Our proposed Explainable and Lightweight Lung Cancer Net addresses the limitations identified in previous works by combining the strengths of CNNs, lightweight architectures, and XAI

techniques. XLLC-Net is specifically designed to classify lung cancer from histopathological images, achieving high performance with minimal computational overhead.

## 3 Methodology

This section outlines the methodology utilized in this study, covering data acquisition, pre-processing techniques, the integration of XLLC-Net, and the evaluation metrics employed to assess the effectiveness of the proposed architecture in classification tasks. Fig 1 illustrates the step-by-step process of the proposed approach.

### 3.1 Data acquisition and pre-processing

Our research utilizes the publicly available dataset LC25000 [30]. A Leica LM190 HD camera attached to an Olympus BX41 microscope was used to capture 250 color photos of each cancer subtype, totaling 1,250 photos without any data augmentation [31]. Subsequently, the 250 samples for each cancer subtype were augmented using techniques such as left and right rotations and horizontal and vertical flips, resulting in a total of 5,000 images. As a result of data augmentation, the dataset now includes 25,000 standard histopathology images, all

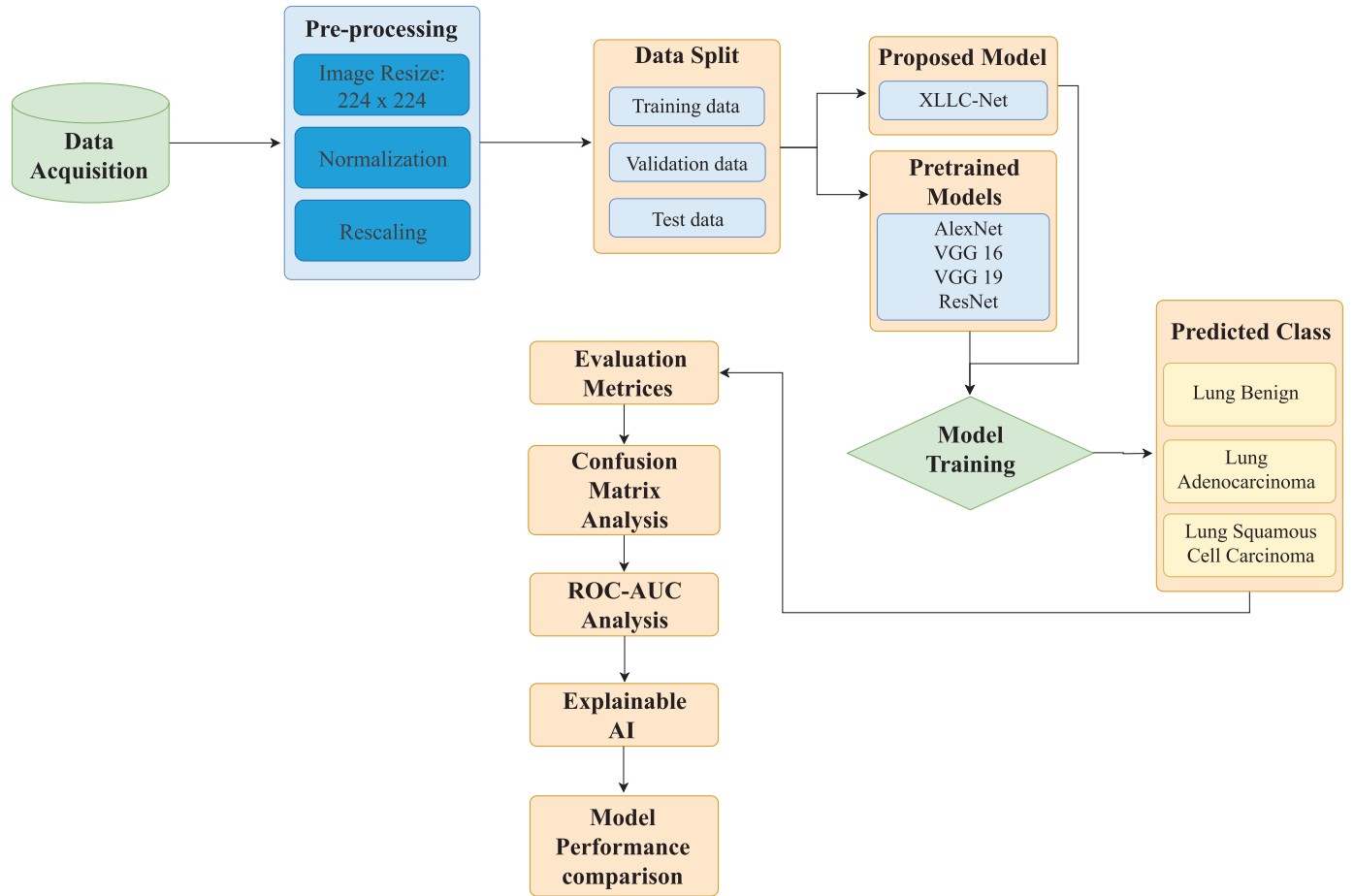

**Fig 1. Illustration of the systematic workflow employed in this study.**

images are 768 x 768 pixels in size and are in jpeg file format, with 10,000 depicting colon cancer and 15,000 depicting lung cancer. The lung cancer dataset is categorized into three cell labels: adenocarcinomas, squamous cell carcinoma, and benign tissue. Meanwhile, the colon cancer dataset is divided into two cell labels: adenocarcinomas and benign tissue. For our study, we exclusively utilized the lung cancer image directory within the dataset. This directory is organized into three subfolders, each containing 5,000 images, corresponding to the three cell labels: adenocarcinomas, squamous cell carcinoma, and benign tissue. Fig 2 provides an illustration of the dataset along with a list of the corresponding class names for each category.

The study incorporates essential preprocessing steps to prepare the image dataset for training and validation. Initially, the dataset is split into training, validation, and test sets in an 80-10-10 ratio. To ensure uniformity in input size, all images are resized to 224 x 224 pixels. Normalization is applied to standardize the pixel values across the dataset, rescaling them to a range of [0, 1]. These techniques enhance dataset diversity, allowing the model to generalize effectively to different object orientations and scenarios.

## 3.2 Proposed XLLC-Net architecture

The proposed model is a deep CNN designed for multi-class classification. The architecture consists of several convolutional layers followed by fully connected layers, each serving a specific purpose in the feature extraction and classification process.

### Convolutional layers

The model begins with an input layer designed to handle images of size $224 \times 224$ with three color channels (RGB). The first layer is a convolutional layer with 64 filters, each of size $3 \times 3$, and uses the ReLU activation function. This layer performs the convolution operation, which can be represented mathematically as:

$$Y^{(1)} = f(Wg^{(1)} * X + bs^{(1)}) \tag{1}$$

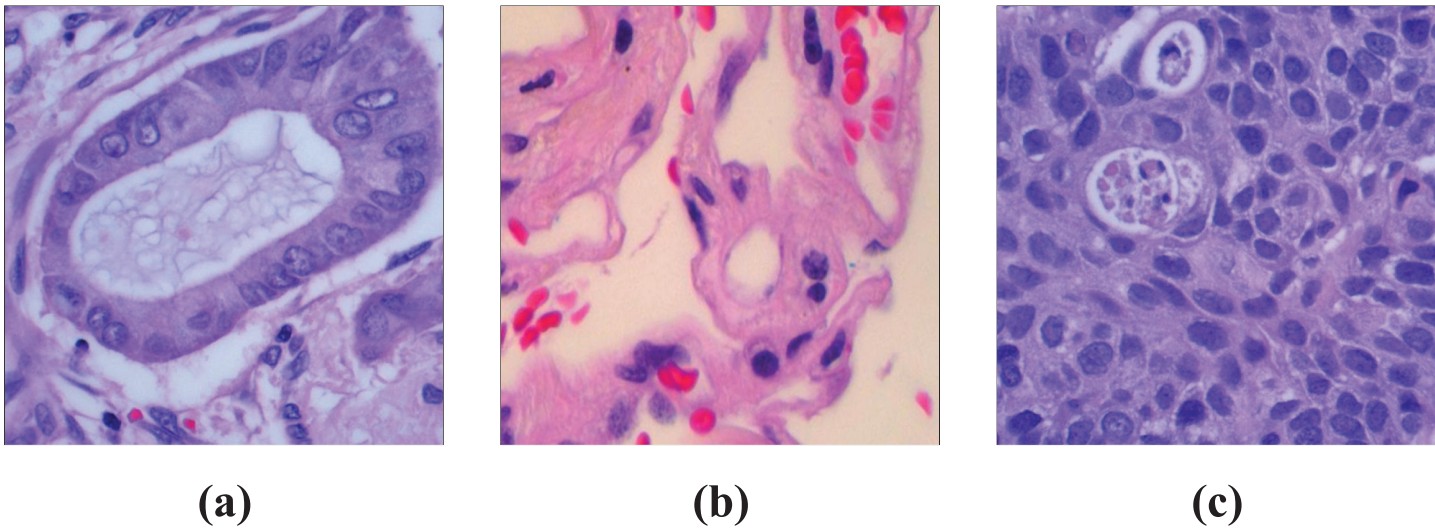

(a) (b) (c)

**Fig 2. The LC25000 dataset where (a) adenocarcinoma, (b) benign, and (c) squamous cell carcinoma.**

where $Wg^{(1)}$ represents the weights, $bs^{(1)}$ the biases, $X$ the input image, and $f$ is the ReLU activation function. Here, the $*$ denotes the convolution operation. The ReLU activation function, defined as:

$$f(z) = \max(0, z) \tag{2}$$

introduces non-linearity to the model by zeroing out negative values, ensuring that the network can learn from complex patterns in the data.

Following the convolutional operation, batch normalization is applied to stabilize and accelerate the training process by normalizing the output of the convolutional layer. Next, the model applies a max-pooling layer with a pool size of $2 \times 2$. The max-pooling operation reduces the spatial dimensions of the feature maps by taking the maximum value in each $2 \times 2$ window, which helps in making the model more invariant to small translations. This operation can be expressed as:

$$Y^{(1)}_{\text{pooled}} = \text{maxpool}(Y^{(1)}, 2 \times 2) \tag{3}$$

where maxpool denotes the max-pooling function.

After pooling, a dropout layer with a dropout rate of 0.3 is applied. Dropout randomly sets 30% of the inputs to zero during training to prevent overfitting and improve generalization.

The second convolutional block repeats a similar sequence of operations but with an increased number of filters (128) to capture more complex features. The operations are:

$$Y^{(2)} = f(W^{(2)} * Y^{(1)}_{\text{pooled}} + b^{(2)}) \tag{4}$$

$$Y^{(2)}_{\text{pooled}} = \text{maxpool}(Y^{(2)}, 2 \times 2) \tag{5}$$

The third convolutional block further increases the number of filters to 256, following the same sequence of convolution, batch normalization, max-pooling, and dropout:

$$Y^{(3)} = f(W^{(3)} * Y^{(2)}_{\text{pooled}} + b^{(3)}) \tag{6}$$

$$Y^{(3)}_{\text{pooled}} = \text{maxpool}(Y^{(3)}, 2 \times 2) \tag{7}$$

The fourth convolutional block reduces the number of filters back to 128, again applying convolution, batch normalization, max-pooling, and dropout:

$$Y^{(4)} = f(W^{(4)} * Y^{(3)}_{\text{pooled}} + b^{(4)}) \tag{8}$$

$$Y^{(4)}_{\text{pooled}} = \text{maxpool}(Y^{(4)}, 2 \times 2) \tag{9}$$

## Fully connected layers

After the final convolutional block, the model flattens the output into a one-dimensional vector, preparing it for the fully connected layers. The flattening operation can be denoted as:

$$Y_{\text{flat}} = \text{flatten}(Y^{(4)}_{\text{pooled}}) \tag{10}$$

This flattened vector is then passed to a dense layer with 128 units and ReLU activation. The operation of this dense layer is:

$$Y^{(5)} = f(W^{(5)} \cdot Y_{\text{flat}} + b^{(5)})$$ (11)

Batch normalization and dropout (with a rate of 0.5) are again applied to this layer to ensure stability and prevent overfitting.

## Output layer

Finally, the model includes an output dense layer with three units, corresponding to the three classes, and uses the softmax activation function to output class probabilities. The softmax function is defined as:

$$\hat{y}_i = \frac{e^{z_i}}{\sum_{j=1}^{C} e^{z_j}}$$ (12)

where $z_i$ are the logits from the final dense layer, and $C$ represents the total number of classes in the classification task.

## Optimizer

The model was compiled using the Adam optimizer. The Adam optimizer is an adaptive learning rate optimization algorithm designed specifically for training deep neural networks. The learning rate is one of the most important hyperparameters and is set to 0.001. The update rule for the weights is defined as:

$$\theta_{t+1} = \theta_t - \frac{\alpha}{\sqrt{\hat{v}_t} + \epsilon} \hat{m}_t$$ (13)

where $\theta_t$ are the parameters at iteration $t$, $\alpha$ is the learning rate, $\hat{v}_t$ is the bias-corrected second moment estimate, $\epsilon$ is a small constant to prevent division by zero, and $\hat{m}_t$ is the bias-corrected first moment estimate.

## Loss function

The loss function used was categorical crossentropy, which is suitable for multi-class classification problems. It is defined as:

$$L(y, \hat{y}) = -\sum_{i=1}^{C} y_i \log(\hat{y}_i)$$ (14)

where $y_i$ is the true label, $\hat{y}_i$ is the predicted probability for class $i$, and $C$ is the number of classes. The crossentropy loss measures the performance of a classification model whose output is a probability value between 0 and 1.

Accuracy was used as the evaluation metric, which is the ratio of correctly predicted instances to the total number of instances. This combination ensures that the model can learn complex patterns effectively and generalize well to new data.

The architecture and parameters of the model are summarized in Table 1, and the overall structure is depicted in Fig 3.

**Table 1. Summary of the proposed model, detailing the number of parameters and characteristics of each layer.**

| Layer (type) | Output Shape | Params |
|---|---|---|
| Input | (None, 224, 224, 3) | 0 |
| conv2d (Conv2D) | (None, 222, 222, 64) | 1792 |
| batch_normalization | (None, 222, 222, 64) | 256 |
| max_pooling2d | (None, 111, 111, 64) | 0 |
| dropout | (None, 111, 111, 64) | 0 |
| conv2d_1 | (None, 109, 109, 128) | 73856 |
| batch_normalization_1 | (None, 109, 109, 128) | 512 |
| max_pooling2d_1 | (None, 54, 54, 128) | 0 |
| dropout_1 | (None, 54, 54, 128) | 0 |
| conv2d_2 | (None, 52, 52, 256) | 295168 |
| batch_normalization_2 | (None, 52, 52, 256) | 1024 |
| max_pooling2d_2 | (None, 26, 26, 256) | 0 |
| dropout_2 | (None, 26, 26, 256) | 0 |
| conv2d_3 | (None, 24, 24, 128) | 295040 |
| batch_normalization_3 | (None, 24, 24, 128) | 512 |
| max_pooling2d_3 | (None, 12, 12, 128) | 0 |
| dropout_3 | (None, 12, 12, 128) | 0 |
| flatten | (None, 18432) | 0 |
| dense | (None, 128) | 2359424 |
| batch_normalization_4 | (None, 128) | 512 |
| dropout_4 | (None, 128) | 0 |
| dense_1 | (None, 3) | 387 |

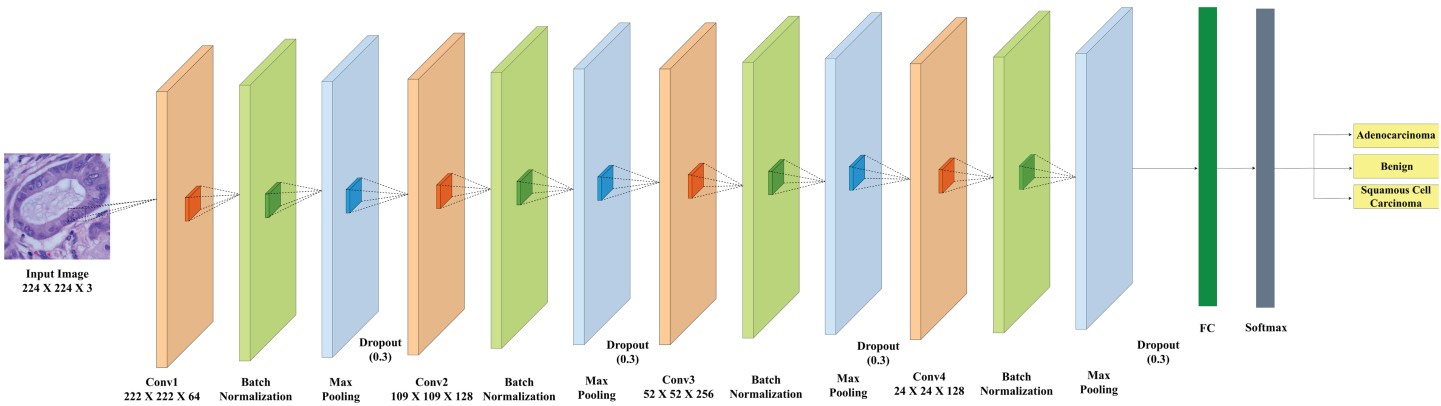

**Fig 3. The proposed XLLC-Net architecture.**

The proposed model was trained for 50 epochs with a batch size of 24. Table 2 lists all the parameters utilized in this study. The Adam optimizer, with a learning rate of 0.001, was employed during training to minimize loss and optimize the model.

## Hyperparameter selection

The selection of hyperparameters for the XLLC-Net model was guided by extensive experimentation and prior literature in DL-based medical imaging. Below is a detailed justification for each key hyperparameter:

- **Number of layers & architecture design:** The proposed XLLC-Net consists of four convolutional layers, followed by a fully connected layer. This depth was chosen to balance

**Table 2. Hyper-parameters employed for training XLLC-Net.**

| Name of the Parameter | Values |
|---|---|
| Batch Size | 24 |
| Epochs | 50 |
| Loss Function | Categorical Cross Entropy |
| Learning Rate | 0.001 |
| Optimizer | Adam |
| Dropout Rate | 0.3 & 0.5 |

computational efficiency and feature extraction. A deeper architecture would increase computational cost, while a shallower network might not capture sufficient image details.

- **Filter sizes and number of filters:** Each convolutional layer uses 3×3 kernel filters, a widely accepted choice for CNN architectures due to their ability to capture local spatial patterns efficiently. The number of filters increases from 64 to 256 in deeper layers, allowing the model to learn hierarchical feature representations.
- **Dropout rates:** Dropout was applied at two stages: 0.3 after convolutional layers and 0.5 before the final dense layer. These values were chosen to mitigate overfitting while ensuring sufficient feature retention. Higher dropout rates (>0.5) led to performance degradation, whereas lower rates (<0.2) resulted in overfitting.
- **Number of epochs:** The model was trained for 50 epochs, as experimental results showed that the validation loss stabilized before this point. Early stopping mechanisms were considered but were unnecessary since model performance did not decline after 50 epochs.
- **Batch size:** A batch size of 24 was used based on GPU memory constraints (NVIDIA RTX 3060Ti). Larger batch sizes (e.g., 32, 64) resulted in memory overflows, while smaller batch sizes (<16) led to longer training times without performance gains.
- **Optimizer & learning rate:** The Adam optimizer was selected due to its adaptive learning rate adjustments, which helped the model converge faster than traditional SGD. A learning rate of 0.001 was empirically chosen, as higher values (e.g., 0.01) caused unstable learning, whereas lower values (e.g., 0.0001) led to slow convergence.
- **Loss function:** Since this is a multi-class classification task, categorical cross-entropy was the optimal choice, as it ensures proper probability distribution learning.

## 3.3 Experimental setup

This section of the paper presents a comprehensive analysis of the suggested method's performance based on the evaluation metrics. The architecture was implemented in Python using Tensorflow and the experiments were conducted using an AMD Ryzen 5 3600 Processor (3.60 GHz), 32 GB RAM, and an Nvidia GeForce RTX 3060ti Graphics Processing Unit (GPU). Table 3 outlines the system specifications on which the proposed work is based.

**Table 3. Specifications of the system used for the proposed framework.**

| Features | Specifications |
|---|---|
| Language | Python (Version - 3.9.16) |
| Environment | Jupyter Notebook |
| Backend | TensorFlow |
| CPU | Ryzen 5 3600 |
| GPU | Nvidia RTX 3060Ti |
| RAM | 32 GB |
| Operating System | Windows 11 |

# 4 Result analysis

The results analysis section of this study provides a comprehensive evaluation of the Explainable and Lightweight Lung Cancer Net in comparison to several established DL models, including AlexNet, VGG16, VGG19, and ResNet50. This evaluation employs a variety of performance metrics such as accuracy, precision, recall, and F1 score to provide a detailed assessment of each model's effectiveness in classifying lung cancer histopathological images.

**Performance metrics:** To ensure the robustness and stability of the proposed XLLC-Net model, we conducted five independent training trials and evaluated their performance on the test set. The results for each trial, including accuracy, precision, recall, and F1-score, are summarized in Table 4.

The results in Table 4 demonstrate the consistency and robustness of the XLLC-Net model across multiple independent training trials. The low standard deviation values indicate that the model maintains stable performance across different runs, confirming its reliability and reproducibility. The accuracy remains consistently high (99.62% ± 0.16%), with precision, recall, and F1-score all maintaining an average above 99.33%, showing minimal performance fluctuations. These findings reinforce the generalization capability of the model, making it highly suitable for lung cancer classification in histopathological images.

Table 5 presents a detailed performance comparison of various pre-trained DL models like ResNet, AlexNet, VGG 16, and VGG 19 with our proposed XLLC-Net model in predicting three types of cancer: Adenocarcinoma, Benign, and Squamous Cell Carcinoma, using Precision, Recall, and F1-Score metrics. The results shown in this table correspond to a single

**Table 4. Performance metrics (Accuracy, Precision, Recall, and F1-score) for five independent training trials of the XLLC-Net model.**

| Trial | Accuracy (%) | Precision (%) | Recall (%) | F1-Score (%) |
|---|---|---|---|---|
| **1st Trial** | 99.58 | 99.00 | 99.00 | 99.00 |
| **2nd Trial** | 99.38 | 100.00 | 99.67 | 99.83 |
| **3rd Trial** | 99.58 | 100.00 | 99.67 | 99.83 |
| **4th Trial** | 99.79 | 100.00 | 100.00 | 100.00 |
| **5th Trial** | 99.79 | 99.67 | 100.00 | 99.83 |
| **Mean ± Std** | **99.62 ± 0.16** | **99.33 ± 0.30** | **99.67 ± 0.30** | **99.70 ± 0.30** |

**Table 5. Performance comparison of different DL models in predicting cancer types.**

| Models | Predicted Labels | Precision | Recall | F1-Score |
|---|---|---|---|---|
| ResNet | Adenocarcinoma | 0.98 | 0.98 | 0.98 |
| | Benign | 1.00 | 1.00 | 1.00 |
| | Squamous Cell Carcinoma | 0.98 | 0.98 | 0.98 |
| AlexNet | Adenocarcinoma | 0.92 | 0.95 | 0.93 |
| | Benign | 1.00 | 0.99 | 0.99 |
| | Squamous Cell Carcinoma | 0.95 | 0.93 | 0.94 |
| VGG 16 | Adenocarcinoma | 0.88 | 0.94 | 0.91 |
| | Benign | 0.98 | 0.99 | 0.99 |
| | Squamous Cell Carcinoma | 0.96 | 0.88 | 0.92 |
| VGG 19 | Adenocarcinoma | 0.91 | 0.93 | 0.92 |
| | Benign | 0.98 | 0.99 | 0.99 |
| | Squamous Cell Carcinoma | 0.95 | 0.91 | 0.93 |
| **XLLC-Net** | Adenocarcinoma | **0.99** | **0.99** | **0.99** |
| | Benign | **1.00** | **1.00** | **1.00** |
| | Squamous Cell Carcinoma | **0.99** | **0.99** | **0.99** |

representative trial. While multiple independent training trials were conducted (as summarized in Table 4), only one trial is displayed here for direct comparison with other models. The low standard deviation values reported earlier further confirm the robustness and stability of XLLC-Net. The proposed model demonstrates the highest overall performance with near-perfect scores (0.99 to 1.00) across all cancer types and metrics, indicating superior accuracy and reliability. ResNet also shows strong and consistent performance, particularly in detecting Benign cases with perfect scores. Though AlexNet, VGG 16, and VGG 19 exhibit slightly lower and more robust performance, overall XLLC-Net outperforms all these models in this comparison.

Furthermore, Table 6 and Fig 4 provide a comparative performance analysis of the proposed XLLC-Net model against several established DL models, namely AlexNet, VGG16, VGG19, and ResNet50. Table 6 details key performance metrics such as accuracy, precision, recall, F1 score, total parameters, and training time per epoch. The proposed XLLC-Net achieves an outstanding overall accuracy of 99.62% ± 0.16%, significantly higher than AlexNet's 95.41%, VGG16's 93.54%, VGG19's 93.75%, and even surpassing ResNet50's

**Table 6. Performance of the proposed model against various DL models.**

| Model | Accuracy (%) | Precision (%) | Recall (%) | F1-score (%) | Total Params | Training Time (s)/Epoch |
|---|---|---|---|---|---|---|
| AlexNet | 95.41 | 95.66 | 95.66 | 95.33 | 46,759,299 | 70 |
| VGG 16 | 93.54 | 94 | 93.67 | 94 | 134,272,835 | 150 |
| VGG 19 | 93.75 | 94.67 | 94.33 | 94.67 | 139,582,531 | 185 |
| ResNet-50 | 98.54 | 98.67 | 98.67 | 98.67 | 23,593,859 | 95 |
| **Proposed** | **99.62 ± 0.16** | **99.33 ± 0.30** | **99.67 ± 0.30** | **99.70 ± 0.30** | **3,028,483** | **60** |

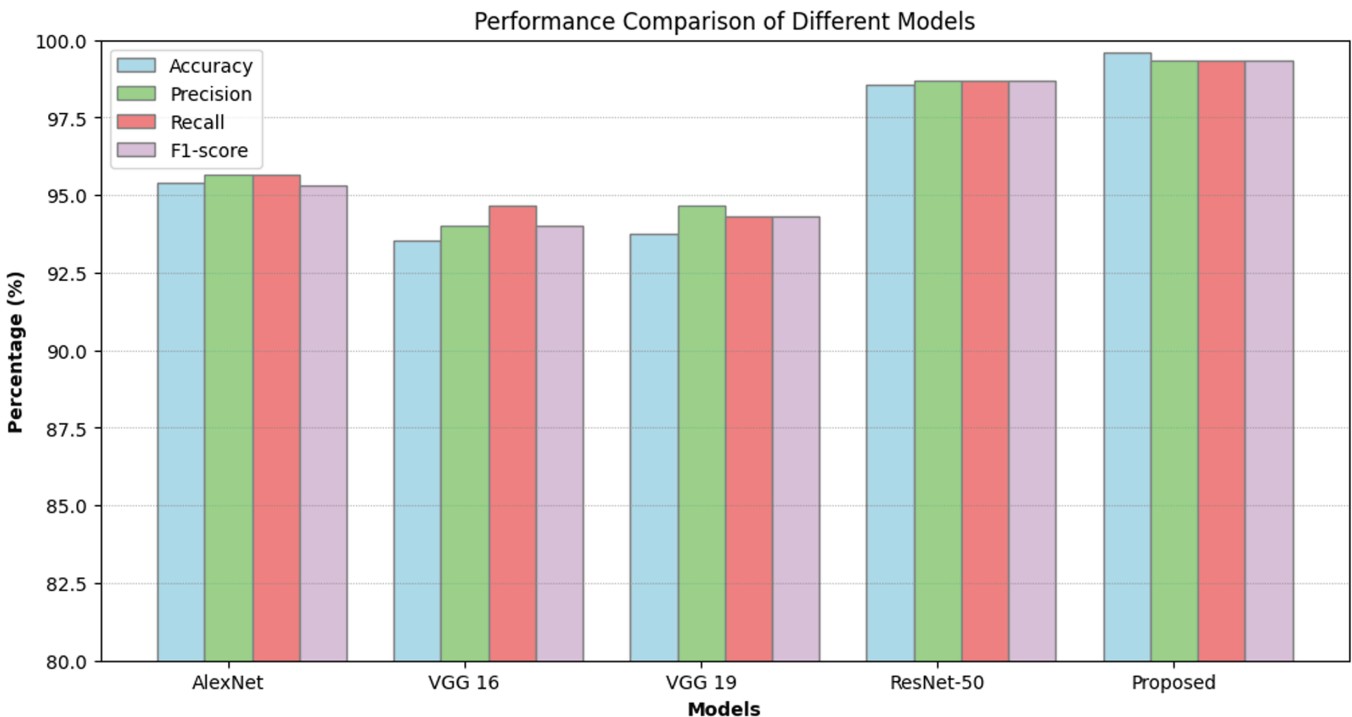

**Fig 4. Performance of the proposed model against various DL models.**

98.54%. XLLC-Net also excels in overall precision, recall, and F1 scores, achieving 99.33% ± 0.30%, 99.67% ± 0.30%, and 99.70% ± 0.30%, respectively, demonstrating its superior classification performance. In terms of computational efficiency, XLLC-Net contains only 3,028,483 parameters and requires just 60 seconds per epoch for training. Moreover, Fig 4 visually reinforces these findings, showcasing the superior performance of XLLC-Net across all metrics, highlighting its efficiency and effectiveness in lung cancer histopathology classification. The bar graph in Fig 4 vividly depicts the performance metrics, emphasizing the model's optimal balance between high accuracy and low computational demands.

Furthermore, Table 7 analyzes the performance comparison of other related research works with our proposed model. First of all, Kumar et al. [32] used CRCCN-Net and achieved an accuracy of 98.30%, with a precision of 97.84%, recall of 98.16%, and an F1-score of 97.99%. This model does not incorporate XAI. In another work, Halder et al. [33] utilized MorphAttnNet, reaching an accuracy of 98.96%, precision of 99.12%, recall of 98.32%, and an F1-score of 98.72%, without using XAI. Singh et al. [34] used an Ensemble method, achieving an accuracy of 99.00%, precision of 99.00%, recall of 98.80%, and an F1-score of 98.90%, without incorporating XAI. Additionally, Wadekar et al. [35] employed a Modified VGG 19, achieving an accuracy of 97.73%, precision of 97.00%, recall of 97.00%, and an F1-score of 97.00%, without using XAI. Tian et al. [36] employed a Feature Pyramid Network (FPN) and Squeeze-and-Excitation (SE) modules combined with a Residual Network (ResNet18), achieving an accuracy of 98.84%, precision of 98.27%, recall of 98.85%, and an F1-score of 98.56%, without incorporating XAI. However, the proposed XLLC-Net in this paper demonstrates superior performance with an accuracy of 99.62% ± 0.16%, precision of 99.33% ± 0.30%, recall of 99.67% ± 0.30%, and an F1-score of 99.70% ± 0.30%. This model uniquely incorporates XAI, enhancing interpretability.

In summary, the proposed XLLC-Net model exhibits exceptional performance in classifying lung histopathological data, outperforming other state-of-the-art models across multiple evaluation metrics. The use of XAI in XLLC-Net further distinguishes it by providing transparency and insights into the model's decision-making process. The presented results indicate that XLLC-Net achieves outstanding accuracy and reliability, making it a robust solution for practical applications in medical image analysis.

**Accuracy graph:** The accuracy graphs, shown in Figs 5 and 6, present the training and validation accuracy over 50 epochs for AlexNet, ResNet50, VGG16, VGG19, and the proposed XLLC-Net model. In Fig 5a, AlexNet shows a steady increase in both training and validation accuracy, reaching approximately 0.95 by the final epoch. However, there are noticeable fluctuations, indicating less stable learning. Fig 5b shows ResNet50 achieving near-perfect accuracy early on, maintaining above 0.98 throughout the epochs, demonstrating its robust performance. Figs 5c and 5d for VGG16 and VGG19 depict slower convergence rates and lower final accuracies of around 0.94 and 0.93, respectively. Fig 6 illustrates the accuracy curves across five independent training trials of the proposed XLLC-Net. Each subplot (a–e) represents a

**Table 7. State-of-the-art comparison between proposed model and previous models on lung histopathological data.**

| Ref. | Model | Accuracy (%) | Precision (%) | Recall (%) | F1-score (%) | Total Params (M) | XAI |
|---|---|---|---|---|---|---|---|
| Kumar et al. [32] | CRCCN-Net | 98.30 | 97.84 | 98.16 | 97.99 | 3.76 | ✗ |
| Halder et al. [33] | MorphAttnNet | 98.96 | 99.12 | - | 98.72 | 3.09 | ✗ |
| Singh et al. [34] | Ensemble | 99.00 | 99.00 | 98.80 | 98.80 | N/A | ✗ |
| Wadekar et al. [35] | Modified VGG 19 | 97.73 | 97.00 | 97.00 | 97.00 | N/A | ✗ |
| Tian et al. [36] | FPN + SE + ResNet18 | 98.84 | 98.27 | 98.85 | 98.56 | N/A | ✗ |
| **This paper** | **XLLC-Net** | **99.62 ± 0.16** | **99.33 ± 0.30** | **99.67 ± 0.30** | **99.70 ± 0.30** | **3.03** | ✓ |

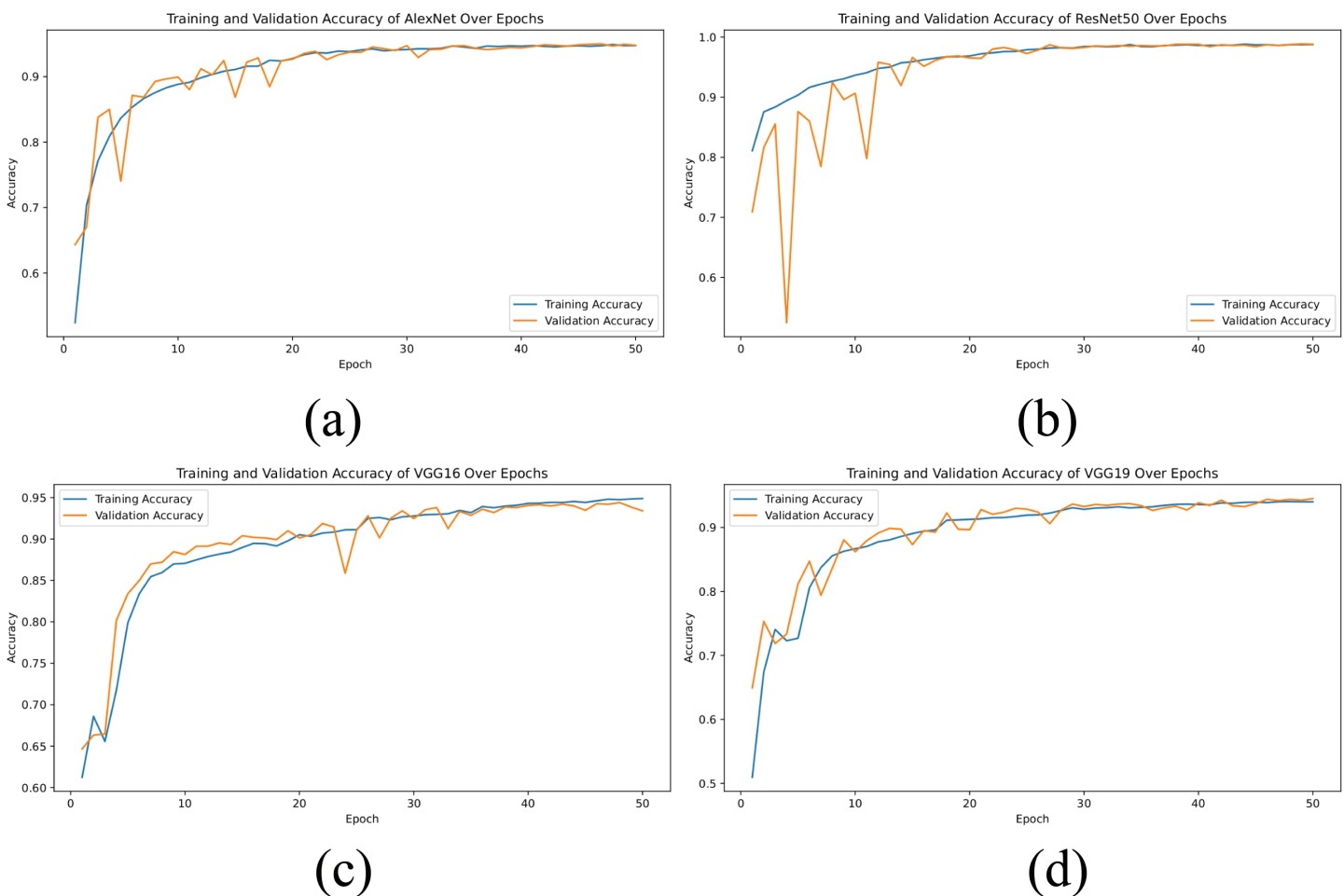

**Fig 5. Comparison of training and validation accuracy across epochs for several pre-trained models, including (a) AlexNet, (b) ResNet50, (c) VGG16, and (d) VGG19.**

separate trial, showing both training and validation accuracy over 50 epochs. Across all trials, XLLC-Net consistently achieves high accuracy, quickly surpassing 0.99 for both training and validation sets. The slight variations in the early epochs indicate normal fluctuations in convergence, yet all trials stabilize towards near-perfect accuracy. These results highlight the robustness and reproducibility of the proposed model, ensuring reliable performance across multiple independent runs. Despite having a significantly reduced computational footprint, XLLC-Net maintains superior generalization and efficiency compared to heavier deep learning architectures.

**Loss graph:** The loss graphs, depicted in Figs 7 and 8, provide insights into the training and validation loss over 50 epochs for the same set of models. In Fig 7a, AlexNet shows a gradual decrease in loss, but with some variability, indicating challenges in achieving stable optimization. Fig 7b for ResNet50 demonstrates a rapid reduction in loss, stabilizing at low values quickly, which correlates with its high accuracy performance. Figs 7c and 7d for VGG16 and VGG19 reveal higher and more fluctuating loss values, suggesting that these models have a harder time optimizing compared to ResNet50. Fig 8 illustrates the training and validation loss curves for five independent trials of the proposed XLLC-Net. Each subplot

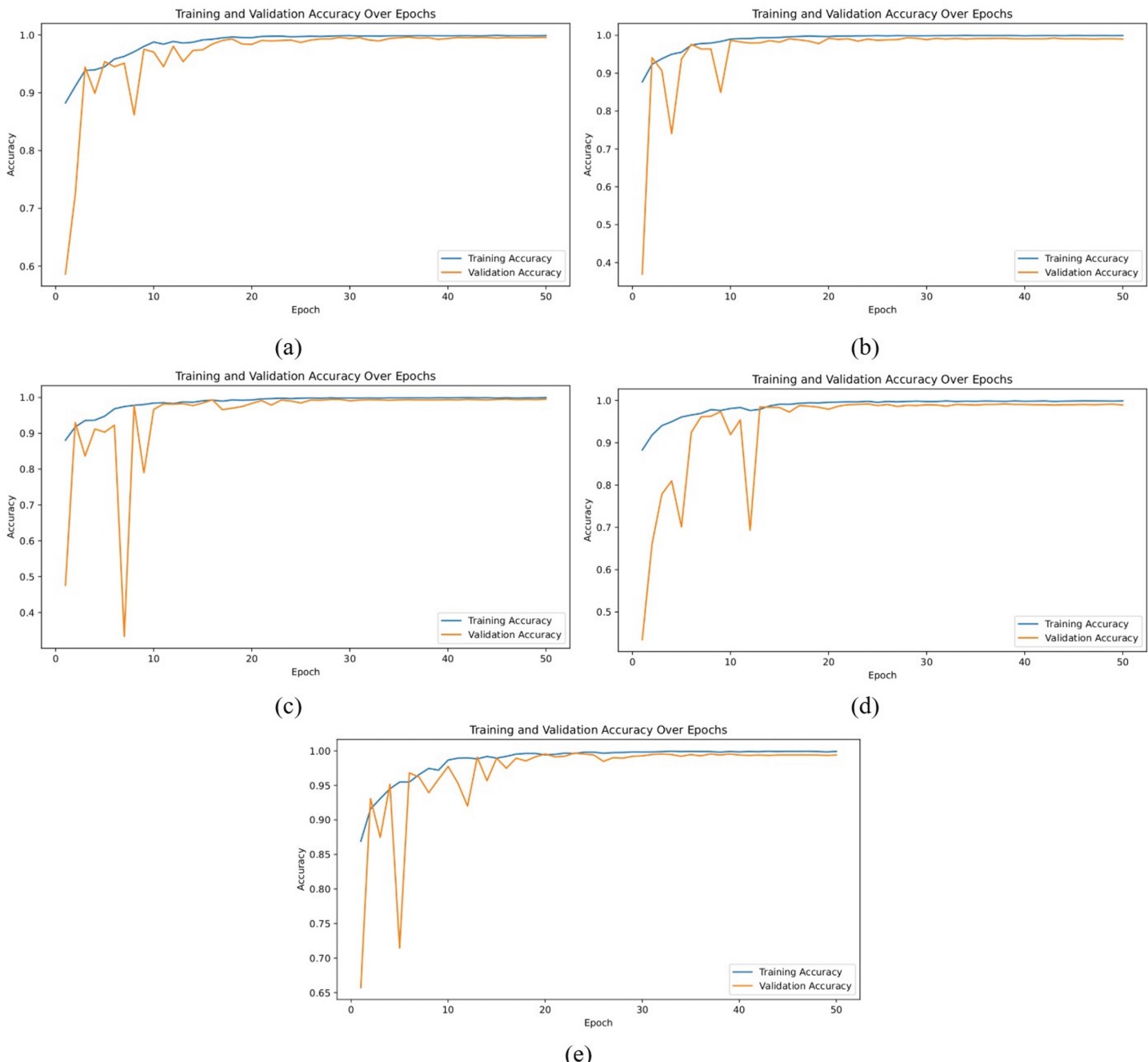

**Fig 6. Training and validation accuracy across epochs for our proposed XLLC-Net model. (a) 1st Trial, (b) 2nd Trial, (c) 3rd Trial, (d) 4th Trial, (e) 5th Trial.**

(a-e) represents a separate training trial, demonstrating the model's consistency across multiple training runs. Across all trials, XLLC-Net exhibits a rapid decline in both training and validation loss, reaching and maintaining minimal loss values early in training. While minor fluctuations appear in some trials, all converge to stable low loss values within the first few epochs. These results further validate the model's robust optimization and efficient learning, ensuring that it generalizes well across different runs while avoiding overfitting.

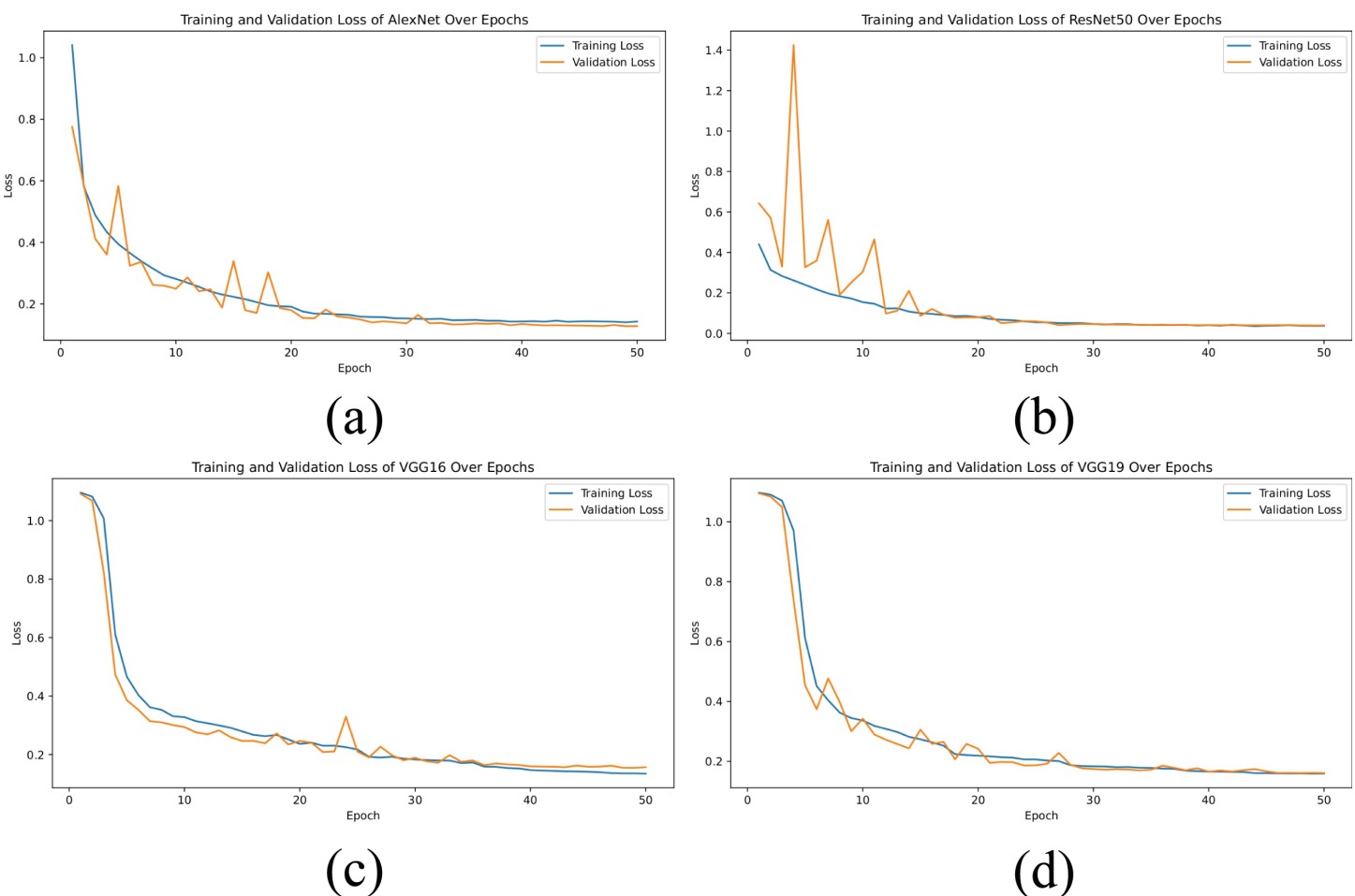

**Fig 7. Comparison of training and validation loss across epochs for several pre-trained models, including (a) AlexNet, (b) ResNet50, (c) VGG16, and (d) VGG19.**

**ROC curves:** Figs 9 and 10 illustrate the Receiver Operating Characteristic (ROC) curves for the various models: AlexNet, ResNet50, VGG16, VGG19, and the proposed XLLC-Net. Each ROC curve plots the true positive rate against the false positive rate for the three classes: adenocarcinoma, benign, and squamous cell carcinoma. In Fig 9a, the ROC curves for AlexNet show strong performance with near-perfect curves for all classes, indicating high discriminative ability. Fig 9b for ResNet50 demonstrates nearly flawless ROC curves, with the curves closely hugging the top left corner of the plot, suggesting excellent classification capability. Figs 9c and 9d for VGG16 and VGG19 also display robust ROC curves, though they do not consistently reach the same level of near-perfection seen in ResNet50. Fig 10 displays the ROC curves for the proposed XLLC-Net across five independent trials. Each subplot (a-e) corresponds to a different trial, all consistently achieving an AUC of 1.00 for all classes. This confirms the model's exceptional classification capability, perfectly distinguishing between cancer types. The results demonstrate XLLC-Net's stability and robustness across multiple training iterations, reinforcing its reliability for lung histopathology classification.

**Confusion matrices:** The confusion matrices for AlexNet, ResNet50, VGG16, VGG19, and the proposed XLLC-Net provide a detailed view of each model's classification performance. In Fig 11a, AlexNet shows a substantial number of misclassifications, particularly with 25

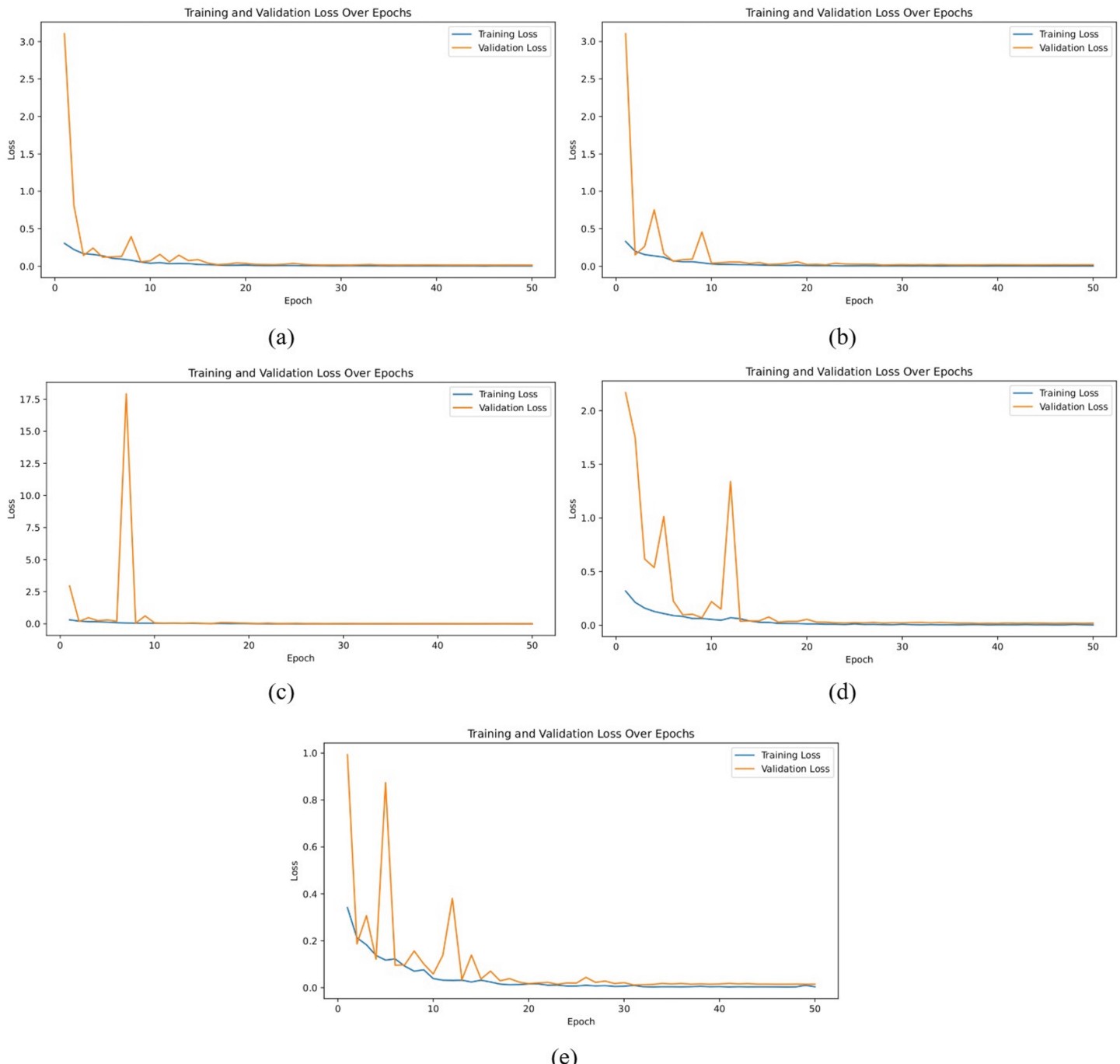

**Fig 8. Training and validation loss across epochs for our proposed XLLC-Net model. (a) 1st Trial, (b) 2nd Trial, (c) 3rd Trial, (d) 4th Trial, (e) 5th Trial.**

instances of adenocarcinoma and 35 instances of squamous cell carcinoma being incorrectly classified. ResNet50, as shown in Fig 11b, performs significantly better, with very few misclassifications: 9 for adenocarcinoma and 9 for squamous cell carcinoma. VGG16's confusion matrix (Fig 11c) indicates 20 misclassified adenocarcinomas and 59 misclassified squamous cell carcinoma cases, revealing a noticeable drop in performance. Similarly, VGG19 (Fig 11d)

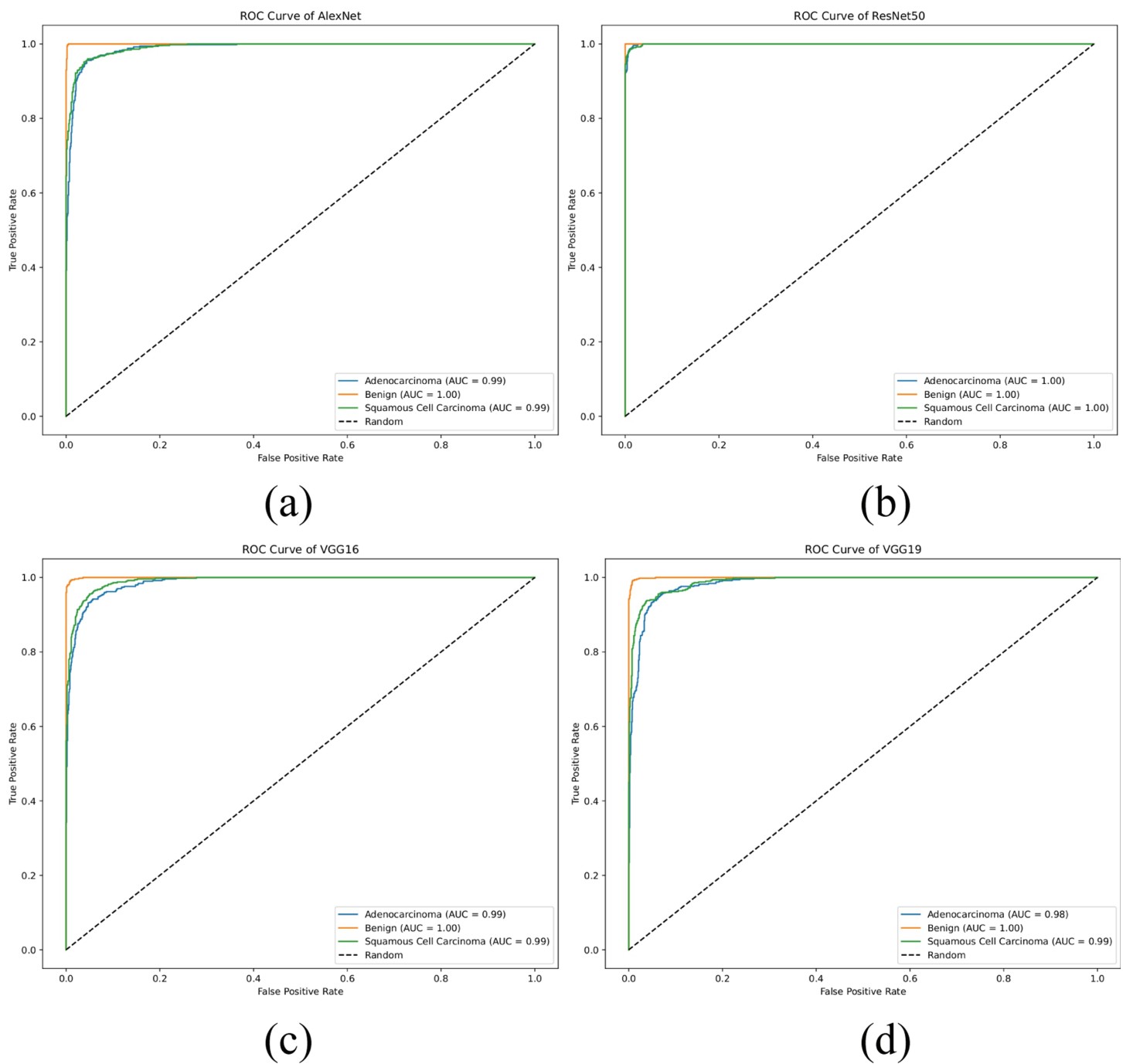

**Fig 9. Comparison of AUC for various pre-trained models, including (a) AlexNet, (b) ResNet50, (c) VGG16, and (d) VGG19.**

shows 26 misclassified adenocarcinoma cases and 44 misclassified squamous cell carcinoma cases. The confusion matrices in Fig 12 illustrate the classification performance of the proposed XLLC-Net model over five independent trials. Each subplot (a–e) corresponds to a separate trial, providing insights into the model's consistency and robustness. Across all trials, XLLC-Net maintains exceptional classification accuracy, with minimal misclassifications. The

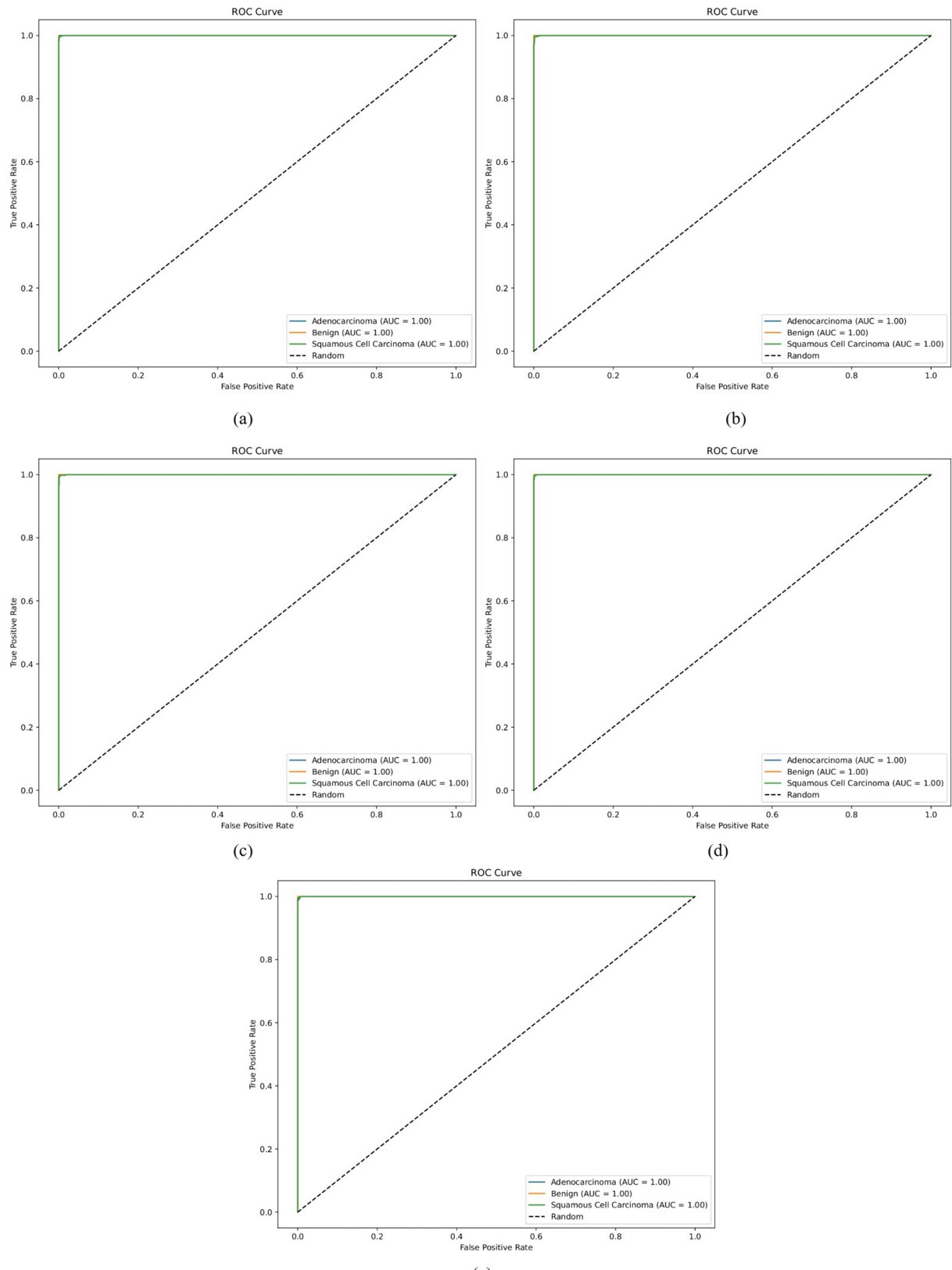

**Fig 10. AUC for our proposed XLLC-Net model across five independent trials. (a) 1st Trial, (b) 2nd Trial, (c) 3rd Trial, (d) 4th Trial, (e) 5th Trial.**

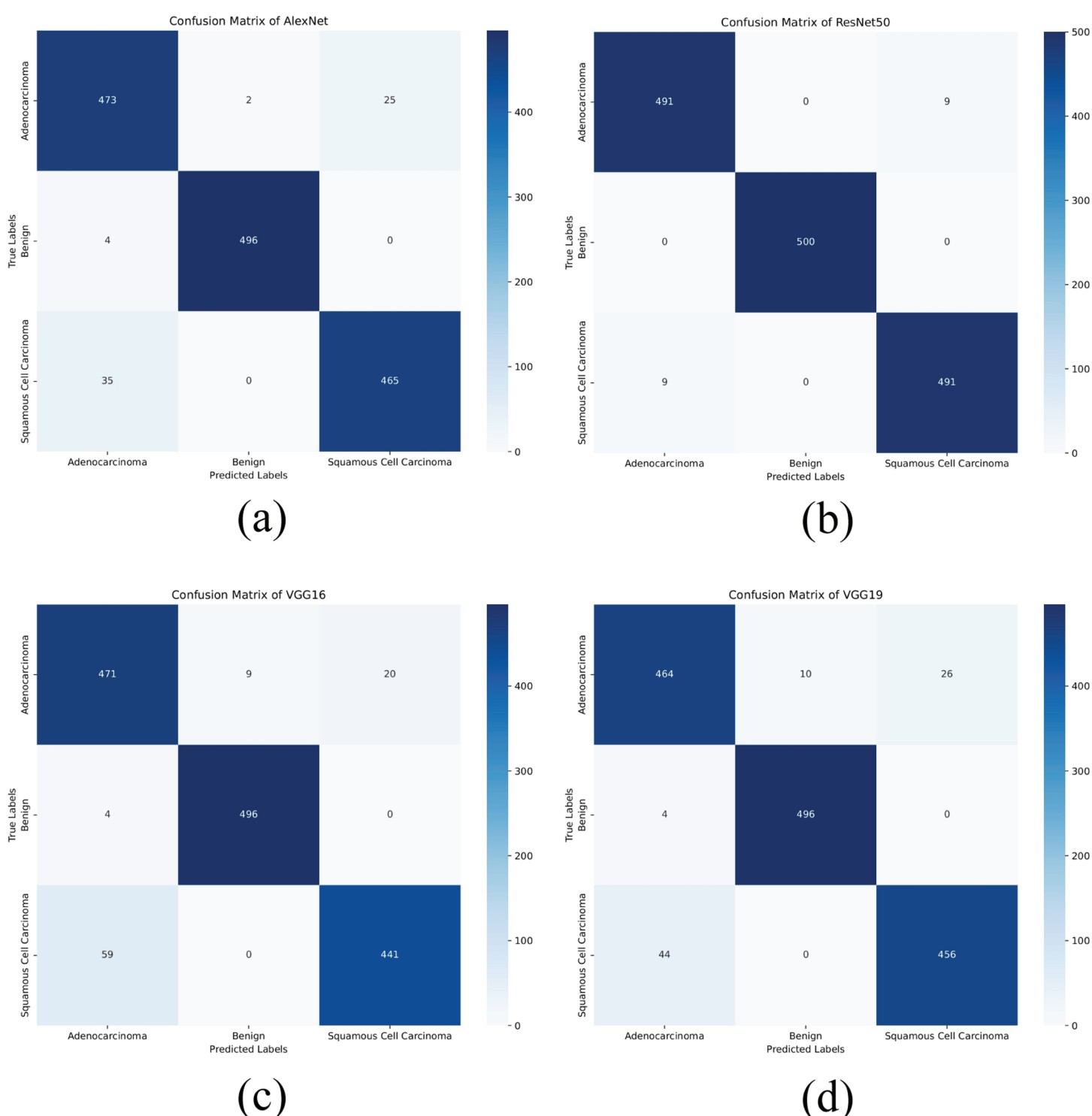

**Fig 11. Comparison of confusion matrix for several pre-trained models, including (a) AlexNet, (b) ResNet50, (c) VGG16, and (d) VGG19.**

model correctly identifies benign cases with perfect accuracy in every trial, while the misclassification rates for adenocarcinoma and squamous cell carcinoma remain extremely low, ranging between 3 to 6 errors per class out of 500 samples.

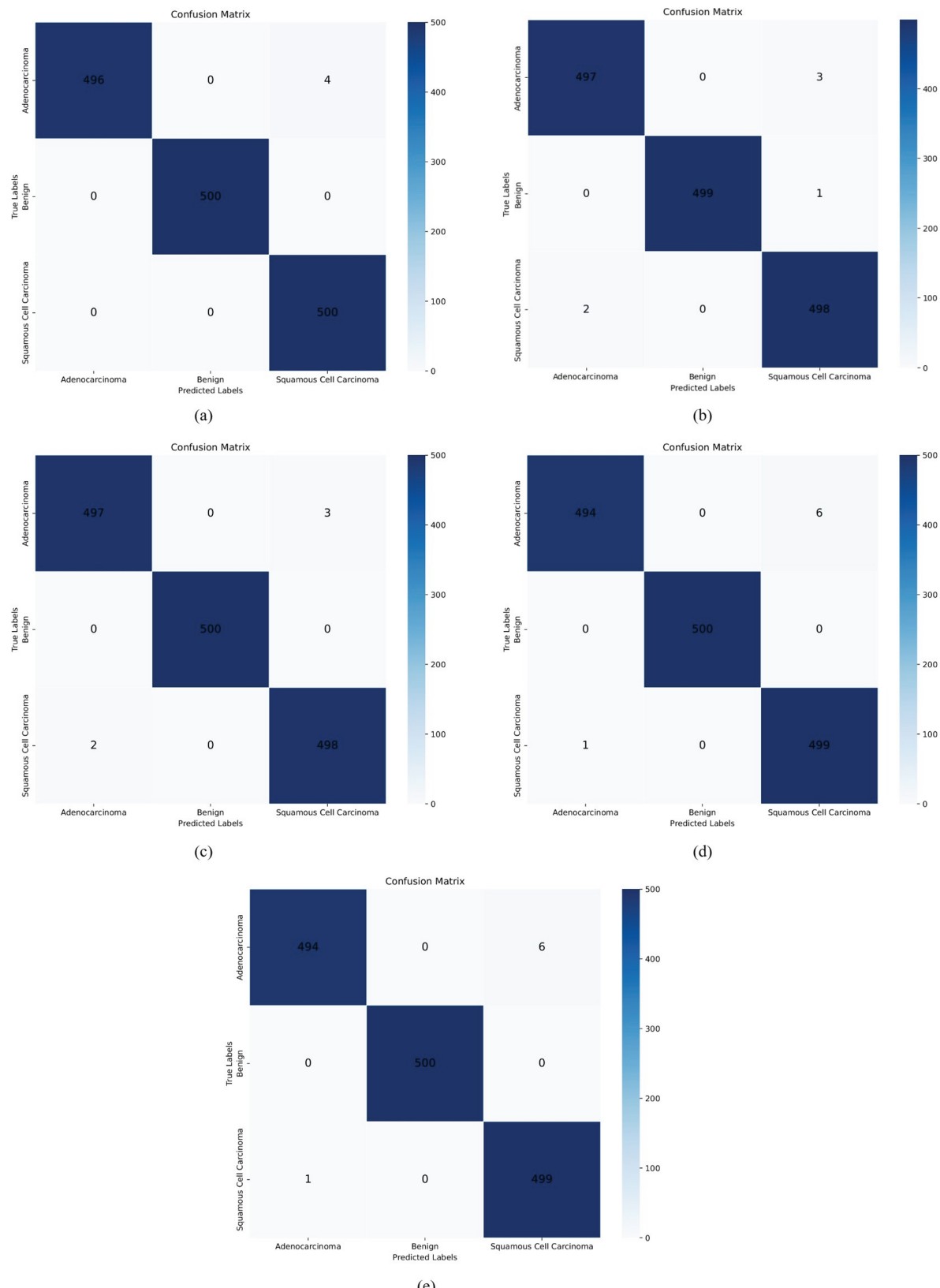

**Fig 12. Confusion matrices of the XLLC-Net model across five independent trials. (a) 1st Trial, (b) 2nd Trial, (c) 3rd Trial, (d) 4th Trial, (e) 5th Trial.**

These results highlight XLLC-Net's high reliability and stability, demonstrating that even across multiple training instances, it consistently maintains outstanding performance in distinguishing different lung cancer types. The minor variations in errors between trials indicate the inherent randomness in model initialization and data augmentation, yet the overall accuracy remains robust, reinforcing XLLC-Net's potential for real-world clinical applications.

**Saliency maps:** Saliency maps, one of the XAI methods used in the paper, highlight the most important pixels in an image that the model focuses on when making a decision. In Figs 13–15, the Saliency Maps for squamous cell carcinoma, benign, and adenocarcinoma, respectively, are displayed. These maps provide a visual representation of the areas that the XLLC-Net model considers critical for its classification. For squamous cell carcinoma (Fig 13), the saliency map shows concentrated areas of high importance, indicating the model's attention to specific cellular structures that are indicative of cancer. Similarly, the saliency map for benign cases (Fig 14) highlights different regions, reflecting the model's

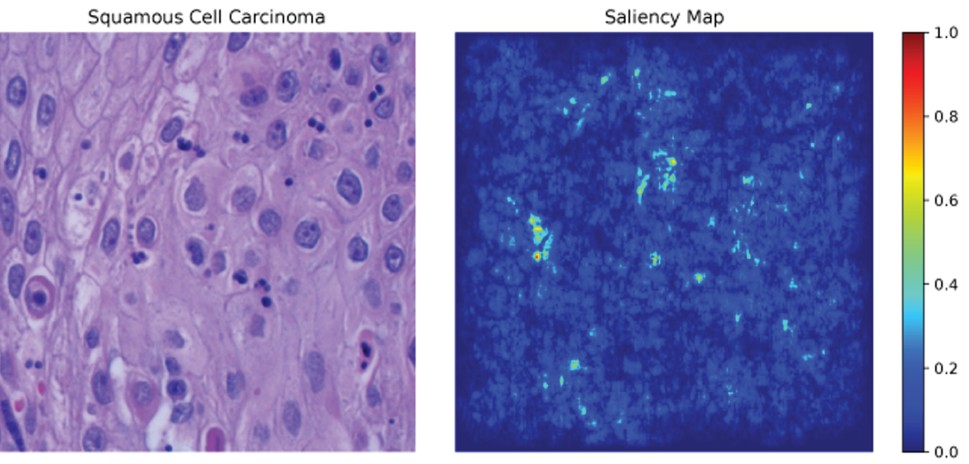

**Fig 13. Saliency map of squamous cell carcinoma.**

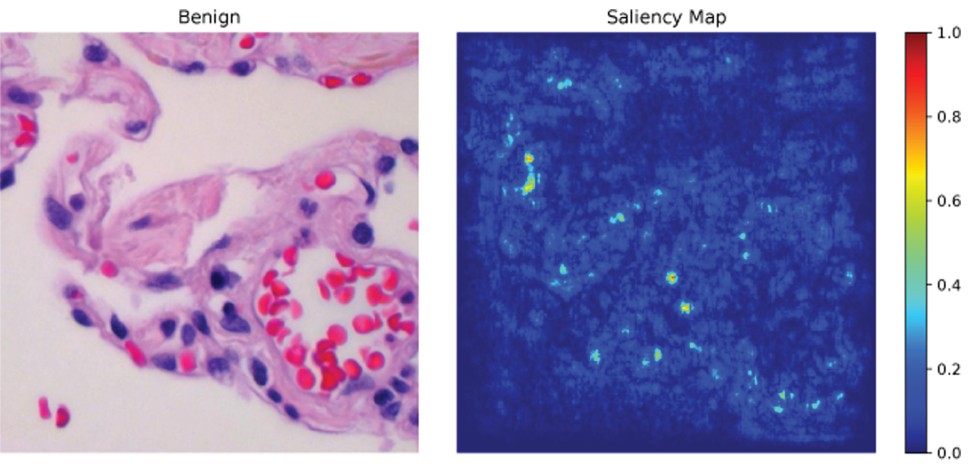

**Fig 14. Saliency map of benign.**

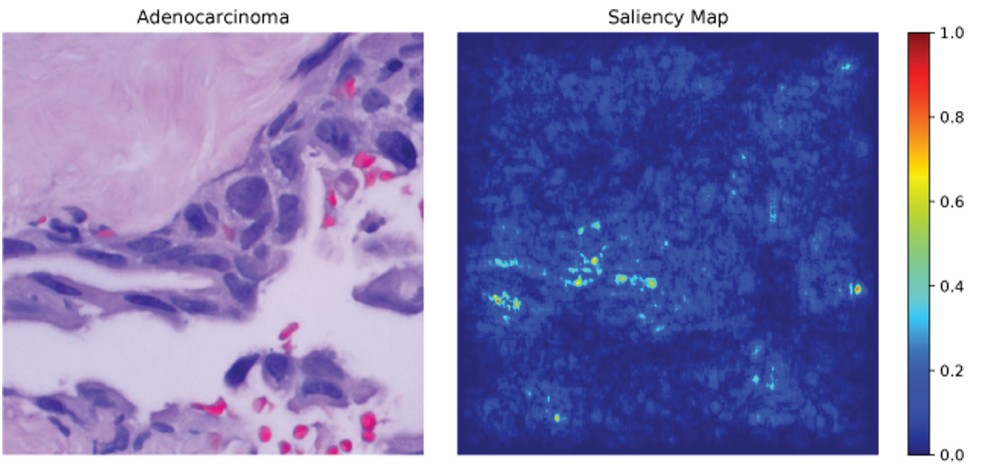

**Fig 15. Saliency map of adenocarcinoma.**

ability to distinguish non-cancerous tissue effectively. The adenocarcinoma saliency map (Fig 15) shows a distinct pattern, with the model focusing on particular regions that are characteristic of this cancer type. These visualizations confirm the model's interpretability and provide insights into its decision-making process, enhancing the trustworthiness of the AI model in clinical settings.

**GRAD-CAM:** GRAD-CAM (Gradient-weighted Class Activation Mapping) is another XAI technique employed in the paper, which generates heatmaps to indicate the areas of the image that are most relevant to the model's classification. Figs 16–18 illustrate the GRAD-CAM results for squamous cell carcinoma, benign, and adenocarcinoma, respectively. In the GRAD-CAM heatmap for squamous cell carcinoma (Fig 16), the highlighted areas show where the model has concentrated its attention, corresponding to significant pathological features. For benign cases (Fig 17), the heatmap indicates different regions, confirming the

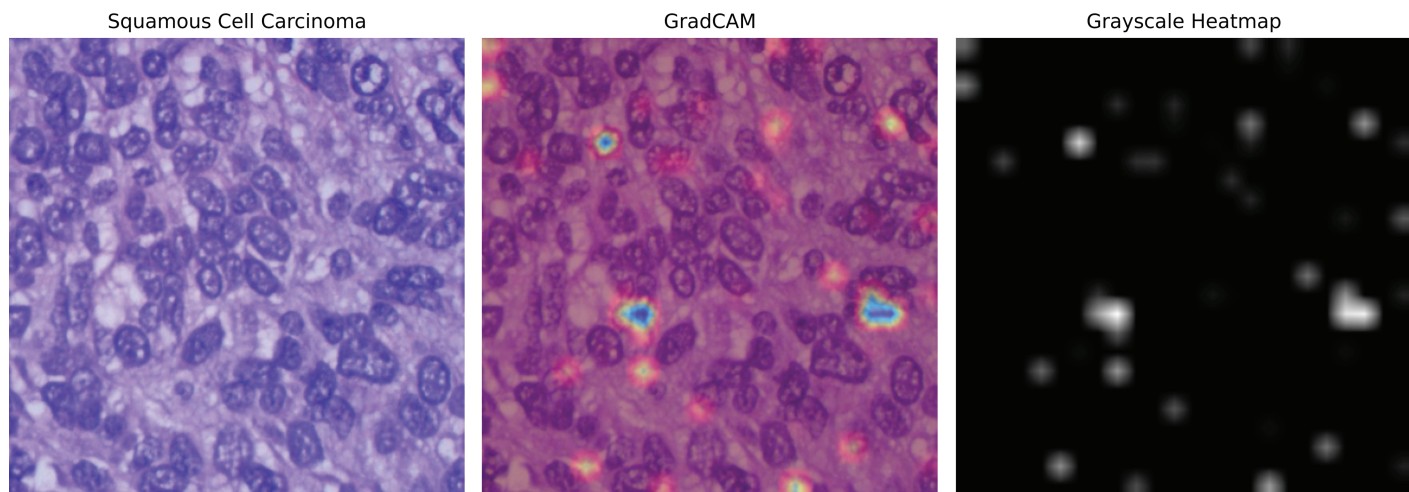

**Fig 16. GradCAM of squamous cell carcinoma.**

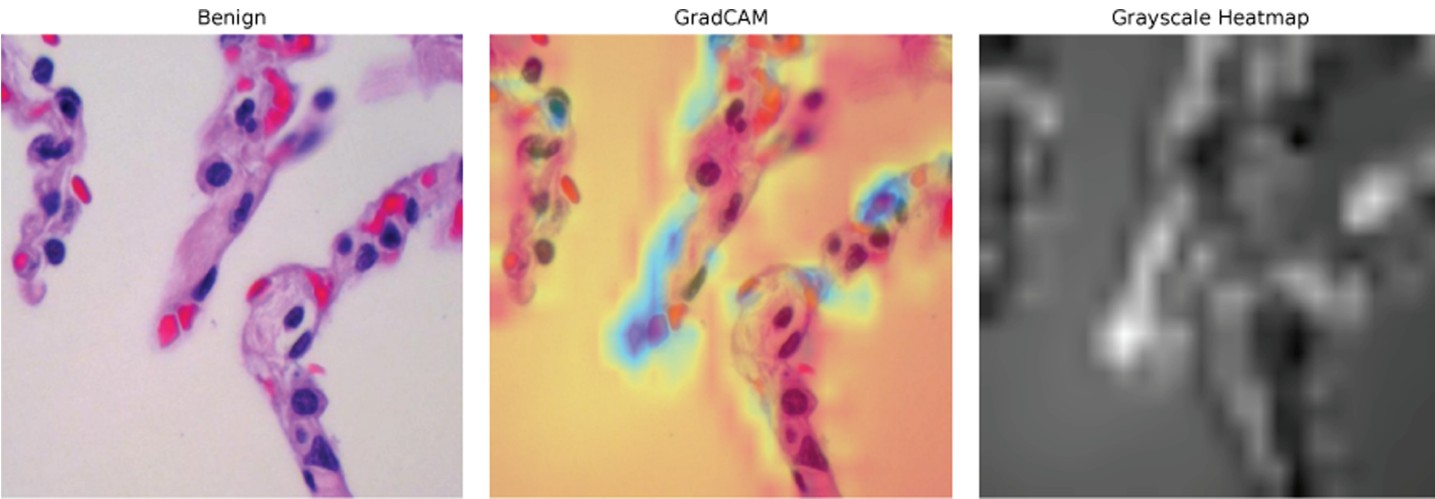

**Fig 17. GradCAM of benign.**

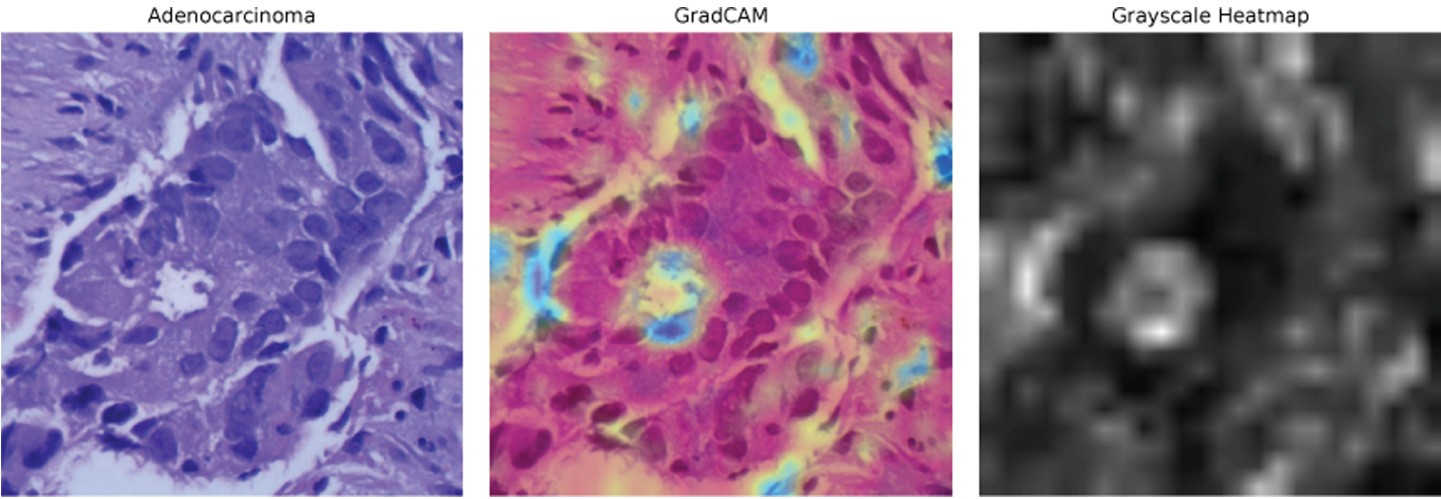

**Fig 18. GradCAM of adenocarcinoma.**

model's capability to identify non-malignant tissue accurately. The adenocarcinoma GRAD-CAM (Fig 18) highlights specific regions that are critical for the model's classification of this cancer type. These heatmaps provide a more intuitive understanding of the model's focus areas, allowing clinicians to visually verify and interpret the AI's decisions. The use of GRAD-CAM thus enhances the transparency and reliability of the XLLC-Net model, supporting its practical application in medical diagnostics.

## 5 Limitations and discussions

While the proposed Explainable and Lightweight Lung Cancer Net (XLLC-Net) exhibits impressive performance with a high classification accuracy of 99.62 ± 0.16, several limitations need to be addressed. Firstly, the model was trained and validated on the

LC25000 dataset, which, although diverse, may not encompass all possible variations in histopathological images encountered in clinical settings. This limitation raises concerns about the generalizability of the model to different datasets and unseen data, potentially affecting its robustness in real-world applications. Moreover, the compact architecture of XLLC-Net, designed to reduce computational overhead, may limit its capacity to learn highly complex patterns compared to larger, more sophisticated models. This constraint could hinder its performance in detecting subtle abnormalities in histopathological images, particularly in cases with overlapping features between cancerous and non-cancerous tissues. Another significant limitation is the dependency on the quality and consistency of the input images. Variations in staining techniques, image resolution, and other preprocessing steps can significantly impact the model's performance. Additionally, while the integration of XAI techniques such as Saliency Maps and GRAD-CAM provides valuable insights into the model's decision-making process, these methods still rely on qualitative visualizations that might not fully capture the complexity of the features influencing the model's predictions. This limitation can lead to challenges in accurately interpreting the model's outputs, particularly in ambiguous cases.

One crucial area of improvement for the XLLC-Net model is the enhancement of its architecture to balance complexity and computational efficiency. Exploring hybrid models that combine lightweight architectures with more complex modules for feature extraction could improve the model's ability to detect subtle patterns without significantly increasing computational requirements. Furthermore, rigorous validation of the model in real-world clinical settings is essential. Collaborating with medical professionals to conduct prospective studies and clinical trials will provide critical feedback on the model's performance and reliability in practice. This real-world validation will help identify and rectify any gaps between the model's performance in controlled experimental settings and its practical utility in diverse clinical environments. By addressing these directions, the XLLC-Net model can be further refined to become a more reliable and widely applicable tool in the early diagnosis and treatment of lung cancer.

## 6 Conclusions

In conclusion, this study underscores the transformative potential of ML and DL in advancing lung cancer diagnosis through histopathological imaging. By introducing the Explainable and Lightweight Lung Cancer Net (XLLC-Net), we have demonstrated the feasibility of achieving both high accuracy and interpretability in DL models for medical image analysis. Our findings highlight the importance of addressing the computational demands and interpretability challenges inherent in DL models to facilitate their practical integration into clinical settings. Through the integration of XAI techniques such as Saliency Maps and GRAD-CAM, XLLC-Net offers transparent insights into its decision-making process, enhancing clinical trust and understanding. Moreover, the lightweight architecture of XLLC-Net not only ensures efficient training but also makes it accessible for deployment in resource-constrained environments. By rigorously validating our model on the LC25000 dataset, encompassing diverse lung and colon cancer classes, we have demonstrated its robustness and potential for real-world application. Overall, this research represents a significant step forward in bridging the gap between sophisticated DL models and their practical utility in medical diagnostics, paving the way for improved patient outcomes and enhanced healthcare delivery.

## Author contributions

**Conceptualization:** Jamin Rahman Jim, Md. Eshmam Rayed.

**Formal analysis:** Jamin Rahman Jim, Md. Eshmam Rayed.

**Methodology:** Jamin Rahman Jim, Md. Eshmam Rayed, M.F. Mridha.

**Resources:** Kamruddin Nur.

**Supervision:** M.F. Mridha.

**Validation:** Kamruddin Nur.

**Visualization:** Md. Eshmam Rayed, Kamruddin Nur.

**Writing – original draft:** Jamin Rahman Jim, Md. Eshmam Rayed.

**Writing – review & editing:** M.F. Mridha, Kamruddin Nur.

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
