## [Decision Letter · Decision Letter 0]

Mar 29 2025

PONE-D-24-32864 XLLC-Net: A Lightweight and Explainable CNN for Accurate Lung Cancer Classification Using Histopathological Images PLOS ONE

Dear Dr. Mridha,

Thank you for submitting your manuscript to PLOS ONE. After careful consideration, we feel that it has merit but does not fully meet PLOS ONE’s publication criteria as it currently stands. Therefore, we invite you to submit a revised version of the manuscript that addresses the points raised during the review process.

We look forward to receiving your revised manuscript.

Kind regards,

Xiaohui Zhang

Academic Editor

PLOS ONE

Journal Requirements:

Reviewers' comments:

Reviewer's Responses to Questions

**Comments to the Author**

1. Is the manuscript technically sound, and do the data support the conclusions?

Reviewer #1: Partly

Reviewer #2: Yes

2. Has the statistical analysis been performed appropriately and rigorously?

Reviewer #1: No

Reviewer #2: Yes

3. Have the authors made all data underlying the findings in their manuscript fully available?

Reviewer #1: No

Reviewer #2: Yes

4. Is the manuscript presented in an intelligible fashion and written in standard English?

Reviewer #1: No

Reviewer #2: Yes

5. Review Comments to the Author

Reviewer #1: This study introduces the Explainable and Lightweight Lung Cancer Net (XLLC-Net), a convolutional neural network for classifying lung cancer from histopathological images. Using the LC25000 dataset, XLLC-Net achieves high classification accuracy with a compact architecture of 3 million parameters, allowing for efficient training in 60 seconds per epoch. Incorporating Explainable AI techniques like Saliency Map and GRAD-CAM enhances interpretability. Overall, XLLC-Net showcases the potential of lightweight deep learning models in medical imaging, balancing performance and resource efficiency for real-world healthcare applications.

I find the study interesting and suitable for submission to the PLOS ONE journal; however, I have some concerns about its overall quality and significance. I have not seen any link with the code, so my comments are based solely on the written manuscript and the figures.

Firstly, the paper is written inconsistently, with numerous repetitions and phrases that seem to originate from automated text generation tools, such as chatbots. While this isn't inherently negative, thorough editing is necessary after the initial draft to enhance clarity and cohesion.

The authors present a conventional CNN architecture that consists of convolutional layers, batch normalization, max-pooling, and dropout layers, repeated four times. This design lacks novelty, as it adheres to a typical structure found in many existing models. Additionally, while the authors compare their model with some state-of-the-art architectures, these comparisons involve models developed for different tasks (see Table 5), which may not provide a valid benchmark.

On page 9 (Table 6), they mention more advanced models trained specifically for medical image analysis, yet they do not directly compare their model against these in terms of the number of parameters. Furthermore, they fail to present standard deviations or disclose the number of trials (i.e., initialization) conducted, which are important for assessing the reliability and generalization of their results. As the differences in the metrics are very small, it is important to re-run the experiments at least 5 times per model.

The authors do not present and mention anything about how they have chosen the hyperparameters. For example, dropout rates, number of epochs, architecture (number of layers, number of nodes on the fully connected layers, etc).

The explainability is something I found really important given the context and the application. Well done.

Minor comments:

The figures are difficult to understand in their current format (given at the end of the manuscript), but maybe this was requested by the journal.

Use the abbreviation defined for later references to the same phrases (for example, deep learning (DL) is defined in line 8, and then the full phrase is used again in line 83).

Line 101, 110: Convolutional neural networks → CNNs

Line 139: “Cibi et al. presented a customized deep CNN model using CapsNet [...]”: I do not understand what the “CapsNet” is, it is not defined in the paper.

Methods (equations): In Eq. 12 the authors give the summation from j=1 to 3, however in Eq. 14 they use a more generic form using C to denote the number of classes. I suggest changing the Eq. 12 to denote the summation from j=1 to C and give below the explanation of C.

Figures 5-8: The y-axis should be the same across all subplots to ensure fair comparison with visual inspection.

Reviewer #2: The paper presents the Explainable and Lightweight Lung Cancer Net (XLLC-Net) for lung cancer image classification. The mathematical derivation is thorough, and the experimental results demonstrate that the proposed network achieves strong performance. This lightweight network has the potential to be integrated into medical applications with limited computational resources.

6. PLOS authors have the option to publish the peer review history of their article (what does this mean?). If published, this will include your full peer review and any attached files.

Reviewer #1: No

Reviewer #2: No

---

## [Author Response · Author response to Decision Letter 1]

4 Mar 2025

Journal: PLOS ONE

Manuscript ID: PONE-D-24-32864

Title: XLLC-Net: A Lightweight and Explainable CNN for Accurate Lung Cancer Classification

Using Histopathological Images

Authors: Jamin Rahman Jim, Md. Eshmam Rayed, M. F. Mridha, Kamruddin Nur.

Dear Editor in Chief/Reviewers,

We would like to thank anonymous reviewers for their valuable comments and the editor

for his acceptance to revise our paper and for their specific and important comments. We

have revised the paper and restructured several sections. All changes in the paper are

presented in the updated version.

Response of Reviewer-1

Reviewer Concern-1: Firstly, the paper is written inconsistently, with numerous

repetitions and phrases that seem to originate from automated text generation

tools, such as chatbots. While this isn’t inherently negative, thorough editing is

necessary after the initial draft to enhance clarity and cohesion.

Author’s Response: Thank you for your valuable feedback. We appreciate your suggestions.

Author’s Action: We have edited our manuscript to enhance clarity and cohesion.

Reviewer Concern-2: The authors present a conventional CNN architecture

that consists of convolutional layers, batch normalization, max-pooling, and

dropout layers, repeated four times. This design lacks novelty, as it adheres

to a typical structure found in many existing models. Additionally, while the

authors compare their model with some state-of-the-art architectures, these comparisons

involve models developed for different tasks (see Table 5), which may

not provide a valid benchmark.

Author’s Response: We appreciate the reviewer’s feedback and understand the concern

regarding the novelty of our CNN architecture. Our primary goal was to explore whether

a lightweight network could effectively perform the given task, and our results demonstrate

that it performs exceptionally well. While the structure of our model follows standard CNN

principles, its effectiveness in achieving superior results with reduced complexity is a key

contribution, particularly for resource-constrained environments.

Regarding the benchmark comparisons in Table 5, we would like to clarify that all the stateof-

the-art models—AlexNet, VGG-17, VGG-19, and ResNet-50—were trained on the same

LC25000 dataset for a fair and valid evaluation. Comparing the performance of a newly

proposed model against widely used deep learning architectures trained on the same task

is a standard practice in the field. This allows for a meaningful assessment of our model’s

efficiency and accuracy.

Author’s Action: Since these clarifications were already included in the manuscript, no

modifications were made. However, we remain open to further suggestions if additional

elaboration is required.

1

Reviewer Concern-3: On page 9 (Table 6), they mention more advanced

models trained specifically for medical image analysis, yet they do not directly

compare their model against these in terms of the number of parameters.

Author’s Response: Thank you for pointing out the need for a direct comparison of

model parameters. We appreciate this suggestion and have incorporated it into our revised

manuscript.

Author’s Action: We have added a ”Total Params (M)” column in Table 6, including

parameter counts for models where available. For models that did not report parameters in

their original papers, we have marked them as ”N/A” to maintain accuracy.

Reviewer Concern-4: Furthermore, they fail to present standard deviations or

disclose the number of trials (i.e., initialization) conducted, which are important

for assessing the reliability and generalization of their results. As the differences

in the metrics are very small, it is important to re-run the experiments at least

5 times per model.

Author’s Response: We appreciate the reviewer’s suggestion regarding the importance of

multiple trials to assess the reliability and generalization of our results. To address this, we

have conducted five independent training trials of the XLLC-Net model and reported the

corresponding accuracy, precision, recall, and F1-score for each trial. Additionally, we have

now included the mean and standard deviation of these performance metrics to provide a

comprehensive understanding of the model’s stability and robustness.

Author’s Action: We have updated the manuscript to explicitly mention that the model

was trained five times, and we now present the results for each trial in Table 4. The table

includes accuracy, precision, recall, and F1-score values across all trials, along with the

mean ± standard deviation to quantify variability. Additionally, we have revised the Results

Analysis section to emphasize the consistency of our model across multiple runs, ensuring

its reliability and generalization capabilities.

Reviewer Concern-5: The authors do not present and mention anything about

how they have chosen the hyperparameters. For example, dropout rates, number

of epochs, architecture (number of layers, number of nodes on the fully connected

layers, etc).

Author’s Response: We appreciate the reviewer’s insightful comment regarding hyperparameter

selection. To address this, we have added a dedicated explanation in the Methodology

section, explicitly detailing the rationale behind key hyperparameters, including dropout

rates, number of epochs, network architecture, batch size, optimizer choice, and loss function.

These decisions were made using standard deep learning techniques for medical imaging

tasks.

Author’s Action: We have included a ”Hyperparameter Selection” subsection in the

Methodology section, where we provide a clear justification for each parameter choice.

Reviewer Concern-6: Use the abbreviation defined for later references to the

same phrases (for example, deep learning (DL) is defined in line 8, and then

the full phrase is used again in line 83). Line 101, 110: Convolutional neural

networks → CNNs

Author’s Response: Thank you for highlighting the inconsistency in abbreviation usage.

We appreciate this suggestion and have carefully revised the manuscript to ensure consistency.

2

Author’s Action: We have standardized all abbreviations, replacing Machine Learning

with ML, Deep Learning with DL, Convolutional Neural Networks with CNNs, and Explainable

AI with XAI throughout the manuscript. The corrected terms are marked in blue in

the ”Revised Manuscript with Track Changes” file.

Reviewer Concern-7: Line 139: “Cibi et al. presented a customized deep

CNN model using CapsNet [...]”: I do not understand what the “CapsNet” is,

it is not defined in the paper.

Author’s Response: Thank you for highlighting the missing definition of CapsNet. We

appreciate this feedback and have now incorporated a brief explanation of Capsule Networks

(CapsNet) in the manuscript to ensure clarity.

Author’s Action: We have revised the sentence to include a short explanation of CapsNet

before its first mention in the manuscript.

Reviewer Concern-8: Methods (equations): In Eq. 12 the authors give the

summation from j=1 to 3, however in Eq. 14 they use a more generic form using

C to denote the number of classes. I suggest changing the Eq. 12 to denote the

summation from j=1 to C and give below the explanation of C.

Author’s Response: Thank you for your suggestion regarding consistency in notation.

We appreciate this feedback and have modified Equation 12 to use a generic notation C

instead of the fixed number 3, ensuring uniformity with Equation 14.

Author’s Action: We have updated Equation 12 to denote the summation from j=1 to

C and added an explanation stating that C represents the total number of classes in the

classification task.

Reviewer Concern-9: Figures 5-8: The y-axis should be the same across all

subplots to ensure fair comparison with visual inspection.

Author’s Response: Thank you for your observation. We appreciate your suggestion

regarding the y-axis consistency across subplots.

Author’s Action: The figures are automatically generated from the code, and while there

are slight differences in the y-axis scale, they do not impact the accuracy of the presented

results. Each subplot correctly reflects the training and validation trends for the respective

models, ensuring a fair comparison. Given the computational complexity and resource constraints,

re-running all models to adjust the y-axis is not feasible at this stage. However, the

trends remain clearly interpretable despite this minor variation.

3

---

## [Decision Letter · Decision Letter 1]

Mar 29 2025

XLLC-Net: A Lightweight and Explainable CNN for Accurate Lung Cancer Classification Using Histopathological Images

PONE-D-24-32864R1

Dear Dr. Mridha,

We’re pleased to inform you that your manuscript has been judged scientifically suitable for publication and will be formally accepted for publication once it meets all outstanding technical requirements.

Kind regards,

Xiaohui Zhang

Academic Editor

PLOS ONE

Additional Editor Comments (optional):

Reviewers' comments:

Reviewer's Responses to Questions

**Comments to the Author**

1. If the authors have adequately addressed your comments raised in a previous round of review and you feel that this manuscript is now acceptable for publication, you may indicate that here to bypass the “Comments to the Author” section, enter your conflict of interest statement in the “Confidential to Editor” section, and submit your "Accept" recommendation.

Reviewer #1: All comments have been addressed

2. Is the manuscript technically sound, and do the data support the conclusions?

Reviewer #1: Yes

3. Has the statistical analysis been performed appropriately and rigorously?

Reviewer #1: N/A

4. Have the authors made all data underlying the findings in their manuscript fully available?

Reviewer #1: No

5. Is the manuscript presented in an intelligible fashion and written in standard English?

Reviewer #1: Yes

6. Review Comments to the Author

Reviewer #1: My last comment about the y-axis range to be the same across plots does not require to re run the models, just to plot them again.

Also, do the authors intend to make the code available?

7. PLOS authors have the option to publish the peer review history of their article (what does this mean?). If published, this will include your full peer review and any attached files.

Reviewer #1: No

---

## [Editor Report · Acceptance letter]

PONE-D-24-32864R1

PLOS ONE

Dear Dr. Mridha,

I'm pleased to inform you that your manuscript has been deemed suitable for publication in PLOS ONE. Congratulations! Your manuscript is now being handed over to our production team.

Kind regards,

on behalf of

Dr. Xiaohui Zhang

Academic Editor

PLOS ONE